# Time to recovery from necrotizing enterocolitis and its predictors among neonates admitted to Neonatal Intensive Care Unit in Bahir Dar, Ethiopia: A retrospective follow up study, 2022

**Birtukan Ayana Tefera[1]☯\*, Abdurahman Mohammed Ahmed[2]☯\*, Sisay Shewasinad Yehualashet☯[3]☯\***

**1** Neonatal Intensive Care Unit, Felege Hiwot Comprehensive Specialized Hospital, Bahir Dar, Amhara Regional State, Ethiopia, **2** School of Public Health, Asrat Woldeyes Health Science Campus, Debre Berhan University, Debre Berhan, Amhara Regional State, Ethiopia, **3** Department of Paediatrics and Child Health Nursing, School of Nursing and Midwifery, Asrat Woldeyes Health Science Campus, Debre Berhan University, Debre Berhan, Amhara Regional State, Ethiopia

☯ These authors contributed equally to this work.
\* sisyehu@gmail.com (SSY); birtukana681@gmail.com (BAT); der1286@gmail.com (AMA)

**Data Availability Statement:** All relevant data are within the manuscript and its Supporting Information files.

## Abstract

### Background

Necrotizing enterocolitis is one of the most common, life-threatening, gastrointestinal disorders in neonates. The recovery time for neonates with NEC varies depending on disease severity, prompt diagnosis, and effective treatment. Therefore, this study was intended to assess the time to recover from necrotizing enterocolitis and its' predictors among neonates admitted to Neonatal Intensive Care Unit in Bahir Dar City, Ethiopia.

### Methods

An institution-based retrospective follow-up study design was employed. A sample of 361 medical records of neonates with necrotizing enterocolitis was selected using systematic random sampling. Diagnosis of NEC in this study required clinical, laboratory and radiographic findings. The survival function was described using Kaplan Meier survival curve and log-rank test. Bivariate and multivariate Cox-proportional hazard (Cox-PH) regression models were used for analysis.

### Results

The median recovery time from necrotizing enterocolitis for neonates in the neonatal intensive care unit was 12 days. The multivariable Cox-PH model showed that neonates classified as Stage III NEC (AHR: 0.42, 95% CI = 0.23–0.77) and those exposed to perinatal asphyxia (AHR: 0.51, 95% CI: 0.35–0.74) had a negative impact on NEC recovery time. However, neonates with a birth weight of 1500-2499gm (AHR: 1.65, 95% CI: 1.05–2.58)

**Funding:** The authors received no specific funding for this work.

**Competing interests:** The authors have declared that no competing interests exist.

and a platelet count greater than 150,000 (AHR: 1.75, 95% CI: 1.24–2.48) had a positive effect on NEC recovery time.

## Conclusion

The recovery time for neonates in the neonatal intensive care unit with necrotizing enterocolitis was longer. Comorbidities and advanced stage of NEC were associated with prolonged recovery time from NEC. However, neonates with better platelet count and birth weight greater than 1500mg had shorter recovery time from NEC.

## Introduction

Necrotizing enterocolitis (NEC) is one of the most common devastating, life-threatening, gastrointestinal disorder in neonates [1, 2]. Non-specific symptoms of NEC include bloody stools, apneas, bradycardia, abdominal distension, feeding intolerance, and temperature instability [3].

NEC is a leading cause of neonatal morbidity and mortality worldwide, affecting both developed and developing countries [4]. NEC is a common cause of illness in preterm, very low birth weight, and asphyxiated neonates [5]. The incidence of NEC is estimated to be 6% to 10% in neonates with a birth weight less than 1500 grams [6, 7].

The median survival time for low birth weight neonates with NEC was 3 days [8]. The overall death rate for neonates with NEC classified as Bell stage two or higher was 23.5% [4]. The presence of co-morbid conditions may prolong the recovery of neonates from NEC [9]. The cost of NEC staying is significant, representing around 20% of annual NICU expenses. NEC survivors who undergo surgery often have NICU stays that extend beyond 90 days [10]. Prolonged hospitalization and extended parenteral nutrition therapy have been associated with delayed re-feeding in NEC cases complicated by sepsis and edema [11]. According to studies conducted in the Western, recovery from the NEC requires an average of 10 to 14 days in the USA [12] and 14 days in the Netherland [13].

Multiple studies have documented that neonates who develop NEC are at significantly higher risk of neurodevelopmental impairment compared to control neonates [14]. The two-year follow-up study found that growth impairment at discharge was observed in 61% of low birth weight survivors who underwent surgical intervention for necrotizing enterocolitis (NEC), 56% of those who received medical NEC intervention, and 36% of low birth weight survivors without NEC [15]. (and prospective longitudinal study showed that 15 years followed children with NEC had lower weight z-score, lower BMI z-score and lower height z-score [16]. Another prospective follow-up study found that 38% of NEC survivors exhibited severe neurodevelopmental disability after 18–24 months of follow-up, which also indicating extremely low birth weight survivors of NEC were at markedly increased risk for severe neurodevelopmental disability [17]. A study by Sonntag et al. found that neonates with NEC developed neurodevelopmental delays at 12 and 20 months compared to age-matched controls without NEC [14, 18].

According to Previous study findings, factors such as gestational age, sex, the age of the neonate at diagnosis, and the stage of the NEC was predictor to recover from NEC [19]. Reducing transfusion and assuring early colostrum feeding improve outcome of neonate from NEC [9, 20]. Neonates with low Apgar scores, requiring resuscitation, Continuous Positive Airway Pressure (CPAP) ventilation, and mixed feeding had longer recovery time compared to those who hadn't experienced these conditions [19]. The incidence of NEC varies between high-

income and low-income countries, with global rates reported to range from 1% to 7% of all NICU admissions [21].

Limited data is available on the incidence of NEC in Africa. A study conducted in Ethiopia reported the incidence of NEC was 9.7% [22]. These variations highlight the need for more comprehensive research on NEC in Africa [23]. The study conducted in Bahir Dar revealed that the occurrence of NEC after birth typically happened within 1 to 7 days of neonatal life [24]. Despite extensive research on NEC and its risk factors, no studies have investigated the duration of recovery from NEC. Therefore, this study aimed to determine the median time to recovery from NEC and identify predictors of recovery time in the NICU at Bahir Dar Town, Ethiopia. The findings of this study will provide valuable evidence for policy makers, program managers, and healthcare professionals, assisting in making informed decisions regarding NEC care. The results will contribute to preventing and improving outcomes of NEC. It also provides information about the duration of NICU stay for neonates with NEC.

## Methods and materials

### Study design and period

Institution-based retrospective follow-up study was conducted among NEC neonates from July 1st to July 26th, 2022.

### Study settings

This study was conducted in Public Hospitals located in Bahir Dar Town, Amhara Region, Ethiopia. The town is situated 563km northwest of Addis Ababa. The tow has three public hospitals: Felege Hiwot Comprehensive Specialized Hospital (FHCSH), Tibebe Gion Specialized Teaching Hospital (TGSTH), and Addisalem Hospital (AH). Presently, these hospitals provide a wide range of inpatient and outpatient medical treatments.

### Source and study population

The source population included all neonates with NEC who were admitted to the NICU. The study population consisted of all neonates with NEC who were admitted to the NICU from May 2019 to May 2022. Neonates with missing major variables (such as date of diagnosis and date of reaching full enteral feeding) were excluded from the study.

### Sample size determination and procedure

Sample size was determined using single population proportion formula. Considering the following assumption: standard normal distribution with confidence interval (CI) of 95% (Z = 1.96), 10% for incomplete cards, margin of error (d = 3) and standard deviation of time to reach a full enteral feeding was 28 [25]. Based on those assumptions the calculated sample size was 369.

The study participants were selected from the registration book. The medical records of neonates who were admitted with NEC from May 2019 to May 2022 were selected. The total number of neonates with NEC admitted in NICU of Tibebe Gion, Felege Hiwot and Addisalem were 326, 410 and 116 respectively, with a total number of 852 cases over the past 3 years. The sample sizes for each hospital were assigned proportionally based on their previous three years admission report of each hospital, which was 141 at Tibebe Gion hospital, 178 at Felege Hiwot hospital, and 50 at Addisalem hospital. Then, medical records of the neonates were arranged as per the registration book numbers sequence in each hospital in the specified time period. Finally, systematic random sampling technique with every 2 interval was applied to

select medical record for data extraction after the first document was selected by lottery method.

## Measurements

NEC was defined as an acute disorder of the gastrointestinal in the neonatal period with clinical sign, and radiological feature [26]. The main outcome measured in this study was the recovery time of neonates from NEC. The study included neonates who achieved recovery from NEC within 14 days after diagnosis, starting from admission. The censored in this study included neonates who were referred, died, or self-discharged before reaching full calorie feeding. Recovery time was measured from the admission date to the date of NEC recovery/started full enteral feeding. To be classified as a NEC case, a patient must have at least two of the radiographic findings (Pneumatosis intestinalis, Portal venous gas, Pneumoperitoneum), as well as at least two of the clinical signs and symptoms (Abdominal distension, Feeding intolerance, Bloody stools, Lethargy), and at least one of the laboratory findings (Metabolic acidosis, Thrombocytopenia and Neutropenia) [27–30]. Neonates' birth weight was categorized as very low birth weight (<1500gm), low birth weight (1500-2499gm), normal birth weight (2500-3999gm), and macrosomia (>4000gm) [31]. Early-onset NEC was defined as NEC occurring within the first week ($\leq$ 7 days) of a neonate's life, while late-onset NEC referred to NEC occurring after day 7. Gestational age was categorized as very preterm (28–31 weeks), moderate preterm (32–37 weeks), and term (greater than 37+1 weeks) [31]. Thrombocytopenia is when the platelet count is below 150,000/microL [32, 33]. Apgar (Activity, Pulse, Grimace, Appearance and Respiration) is a quick assessment performed on a baby at 1, 5 and 10th minutes after birth and it has three levels of score: Low APGAR score 0–3, Moderate APGAR score 4–6, Normal APGAR score 7–10 [34].

The data was collected by Kobo toolbox from neonate's medical record using pre-tested tool, which prepared by reviewing different literatures. The checklist was containing three parts socio-demographic characteristics, neonatal characteristics and management option used for treating neonatal with NEC disorder.

## Data quality control

Data was collected by three BSc neonatal nurses who were not affiliated with the study sites. They received one-day training on the study purpose, confidentiality, data collection techniques, and how to properly fill the checklist using the Kobo toolbox. A pre-test was conducted on 5% of the sample size. Relevant information was extracted after reviewing the available data. The follow-up period started upon admission to the NICU and ended on the day of recovery time or censoring. Data completeness was closely monitored on a daily basis.

## Data processing and analysis

Data were collected using the Kobo toolbox and exported to STATA version 14 statistical software packages used for analysis. Each participant's outcome was divided into two categories: recovery times (the event of interest), recorded as "1," and censorship, coded as "0." The data primarily organized and summarized using the descriptive statistics to show the characteristics of study variables. Days were used as a time scale to calculate time to recovery. The outcome of each participant was dichotomized into censored or event. Median was calculated for recovery time. Kaplan Meier survive curve was used to estimate the time to recovery during the follow-up. The life table was used to estimate cumulative recovery time probabilities after admission. The log-rank test was employed to compare statistical differences between groups of independent variables. Cox proportional hazards regression model was used to identify the

## Supplementary files

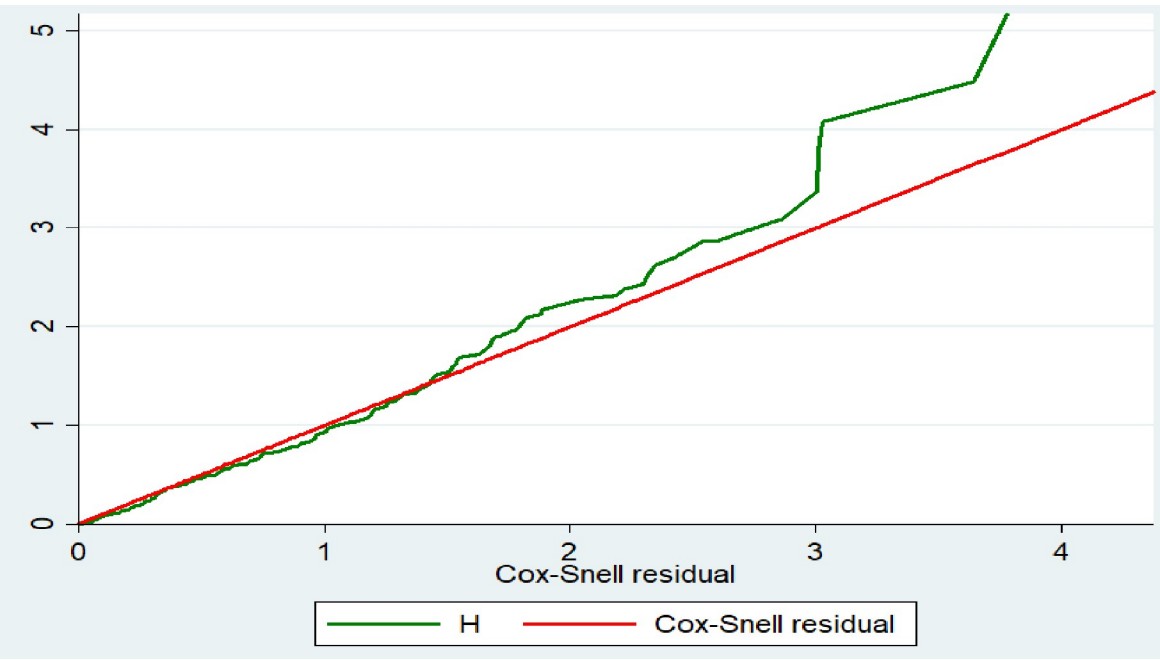

**Fig 1. Cox-Snell residual graph for the goodness of model fitness.**

determinants of recovery time from NEC. All covariates with a p-value of 0.25 or less in the bivariate analysis were entered in to the multivariable model. Adjusted hazard ratios with 95% Confidence Interval (CI) were presented. A p-value less than 0.05 were a cut point to determine statistically significant association between recovery time and covariates. The proportional hazard assumption was checked by using global test and Schoenfeld residual test. The Multicollinearity was checked using Variance Inflation Factor (VIF) the mean result was (1.92).

### Statistical model and assumptions

The Cox proportional hazard model assumption was assessed using the Log-log plot and the Schoenfeld residual (Global) test. The graphical plot of the cumulative hazard versus cox-Snell residuals showed a straight line with a slope of one, indicating that the model fits the data well (Fig 1). Additionally, the final model underwent a statistical fitness test using the global goodness of fit test, which was considered adequate (p-value = 0.4).

### Ethical consideration

Ethical clearance was obtained from Debre Berhan University Aserat Woldeyes Health Science Campus Institution Review Board (No. AWHSC 122/2022). Support letter was written from Amhara public Health Institute (APHI) for each hospital. Formal letter was submitted to FHCSH, TGSTH and AH and permission was assured. Permission was taken from quality improvement offices. All information collected from neonates' cards was kept strictly confidential and anonymous.

## Results

### Socio-demographic characteristics of neonate and mother

In this retrospective follow-up study, 369 neonatal medical records with NEC were reviewed. Eight records were excluded due to incomplete information. Therefore, the study included 361 completed neonatal records (97%). Among them, 192 (53%) were male, and 269 (74.5%) were in the age group of 1–7 days (Table 1).

### Baseline clinical characteristics of Neonate with NEC

The median weight was 1580 grams with (IQR 1350–2530). Median gestational age was 34 weeks (IQR 31-37wks). In this study, among the participants, 342 neonates (94.7%) had an Apgar score greater than five at 5 minutes. The average Apgar score was 7.65 with a standard deviation of ±1 (Table 2).

### Therapeutic management options used to treat the neonates

Three hundred fifty one (97.23%) were treated by medical management, and 303 (83.9%) were feeding breast milk. Most neonates in respiratory failure were assisted by respiratory support, 132 (36.6%) were assisted by intranasal oxygen (INO2) and 131 (36.3%) by Continuous Positive Airway Pressure CPAP (Table 3).

### Recovery rate and median recovery time from NEC

During follow-up time, a total of 361 neonates were followed for 3667 person-days of risk were observed with a minimum and maximum follow-up time of 4 and 14 days, respectively. Of 361 neonates were followed the 1st neonate was recovered at the 4th day of the follow-up. The median follow up time to recover from NEC was 12 days (IQR 10–14). The overall incidence rate of recovery was 60 per 1000 (95% CI: 53, 69) person-days observation. Out of 361 neonates, 223 (61.8%) of them were Survived the rest 138 (38.2%) neonates with NEC were censored, from those 124 (34.3%) neonates died and 10 (2.8%) neonates had left against medical advice and 4 (1.1%) was disappeared without physician consultation. The incidence rate that NEC neonates recovery was 0.0 at first 3 day, 14.5 at 7th day and 18.2 at 14th day per 1000 person-day(Table 4).

### Kaplan-Meier recovery time estimates of neonatal NEC

Kaplan Meier recovery curve was used to estimate the time to recover from NEC during the follow-up period. The overall Kaplan-Meier estimate shows that the probability of not

**Table 1. Socio-demographic characteristics of a neonate with necrotizing enterocolitis admitted from May 2019 to May 2022 at public hospitals in Bahir Dar (n = 361).**

| Variables | Category | Frequency(Percentage) | Event | Censored |
|---|---|---|---|---|
| Age of the neonate(day) | 1–7 days | 258 (71.47%) | 163 | 95 |
| | >7days | 103 (28.53%) | 60 | 43 |
| Age of the mother(year) | 18–34 years | 301 (83.38%) | 188 | 113 |
| | 35–45 years | 60 (16.62%) | 35 | 25 |
| Sex of neonates | Male | 192(53%) | 119 | 73 |
| | Female | 169(47%) | 104 | 65 |
| Place of delivery | Institutional | 355 (98%) | 220 | 135 |
| | Home | 6 (2%) | 3 | 3 |

**Table 2. Characteristics of neonate with NEC who was admitted from May 2019 to May 2022 at public hospitals in Bahir Dar, Ethiopia, 2022 (n = 361).**

| Variables | Category | Frequency(Percentage) | | Survival Status | |
|---|---|---|---|---|---|
| | | | | Recovery | Censored |
| Birth weight | < 1500gm | 149 | (41.3%) | 50 | 99 |
| | 1500-2499gm | 114 | (31.6%) | 90 | 24 |
| | 2500-3999gm | 94 | (26%) | 79 | 15 |
| | >4000gm | 4 | (1.1%) | 4 | - |
| APGAR score in 5th min | <5 | 19 | (5.3%) | 10 | 9 |
| | >5 | 342 | (94.7%) | 213 | 129 |
| Stage of NEC | Stage 1 | 65 | (18%) | 60 | 5 |
| | Stage 2 | 207 | (57.3%) | 149 | 58 |
| | Stage 3 | 89 | (25.7%) | 14 | 75 |
| GA (Gestational age) | 29-31wk | 91 | (25.2%) | 29 | 62 |
| | 32-37wk | 182 | (50.4%) | 130 | 52 |
| | >37wk | 88 | (24.4%) | 64 | 24 |
| Birth Type | Single | 288 | (79.8%) | 193 | 95 |
| | Twin | 67 | (19.6%) | 30 | 37 |
| | Triplet | 6 | (1.7%) | - | 6 |
| Platelet count | < 150,000 | 151 | (41.8%) | 41 | 110 |
| | >150,000 | 210 | (58.2%) | 182 | 28 |
| PNA(Perinatal Asphyxia) | Yes | 105 | (29.1%) | 36 | 69 |
| | No | 256 | (70.9.4%) | 187 | 69 |
| MAS(Meconium Aspiration syndrome) | Yes | 31 | (8.6%) | 18 | 13 |
| | No | 330 | (91.4%) | 205 | 125 |
| Jaundice | Yes | 61 | (16.9%) | 34 | 27 |
| | No | 300 | (83.1%) | 111 | 189 |

**Table 3. Therapeutic management options used to recovery the neonates with necrotizing enterocolitis among neonates admitted from May 2019 to May 2022 at public hospitals in Bahir Dar, in 2022 (n = 361).**

| Variables | Category | Frequency (%) | | Survival Status | |
|---|---|---|---|---|---|
| | | | | Recovery | Censored |
| Type of feeding | Breast milk | 303 | (83.9%) | 189 | 114 |
| | Formula | 44 | (12.2%) | 26 | 18 |
| | Mixed | 14 | (3.9%) | 8 | 6 |
| Type of management | Medical* | 351 | (97.23%) | 220 | 131 |
| | Surgical** | 10 | (2.8%) | 3 | 7 |
| Type of oxygen support | INO2 | 132 | (36.6%) | 99 | 33 |
| | CPAP | 131 | (36.3%) | 41 | 90 |
| | Off | 98 | (27.1%) | 83 | 15 |
| Gastric residual monitoring | Yes | 187 | (51.8%) | 117 | 70 |
| | No | 174 | (48.2%) | 106 | 68 |
| Feeding advancement | 15-20ml | 272 | (75.3%) | 172 | 100 |
| | 30-40ml | 89 | (24.7%) | 51 | 38 |
| Blood transfusion | Yes | 80 | (22.2%) | 32 | 48 |
| | No | 281 | (77.8%) | 191 | 90 |

**\*:** Intravenous fluid resuscitation, Antibiotics, NPO, Gastric decompression, Pain management

**\*\*:** Bowel Resection, Ostomy Creation, Peritoneal Drainage

**Table 4. Over all life table of survival function (n = 361).**

| Time in day | Beg. total | Recovery | lost | Survival | Standard error | 95% Conf. Int. |
|---|---|---|---|---|---|---|
| 2 | 361 | 0 | 2 | 1.0000 | - | - |
| 3 | 359 | 0 | 6 | 1.0000 | - | - |
| 4 | 353 | 1 | 6 | 0.9972 | 0.0028 | 0.9801 0.9996 |
| 5 | 346 | 6 | 16 | 0.9799 | 0.0075 | 0.9583 0.9904 |
| 6 | 324 | 7 | 9 | 0.9587 | 0.0108 | 0.9312 0.9753 |
| 7 | 308 | 10 | 12 | 0.9276 | 0.0143 | 0.8938 0.9509 |
| 8 | 286 | 23 | 15 | 0.8530 | 0.0199 | 0.8091 0.8875 |
| 9 | 248 | 17 | 12 | 0.7945 | 0.0230 | 0.7450 0.8355 |
| 10 | 219 | 39 | 18 | 0.6530 | 0.0279 | 0.5953 0.7046 |
| 11 | 162 | 14 | 6 | 0.5966 | 0.0293 | 0.5367 0.6514 |
| 12 | 142 | 26 | 5 | 0.4874 | 0.0308 | 0.4258 0.5461 |
| 13 | 111 | 19 | 5 | 0.4039 | 0.0309 | 0.3432 0.4638 |
| 14 | 87 | 61 | 26 | 0.1207 | 0.0219 | 0.0820 0.1674 |

recovering from NEC was 1.0 (100%) during the first 3 days of follow-up. This probability then relatively decreased as the follow-up time increased, as can be observed in the Kaplan-Meier curve shown in Fig 2. After the initial 3 days, the non-recovery probability slowly dropped in a stepwise manner as the follow-up time increased from 0 to 14 days. This indicates that during the first 3 days of follow-up, no neonate was observed to have recovered. The median time to recover from NEC was 12 days (IQR 10–14). During the fourth day of hospital stay, only one NEC neonate was recovered (Fig 2).

## Treatment outcome among neonate admitted with NEC in NICU

Two hundred twenty three (61.8%) observation were recovered at the end of follow up time, 138 (38.2%) neonates with NEC were censored. Among the censored cases, 124(34.3%)

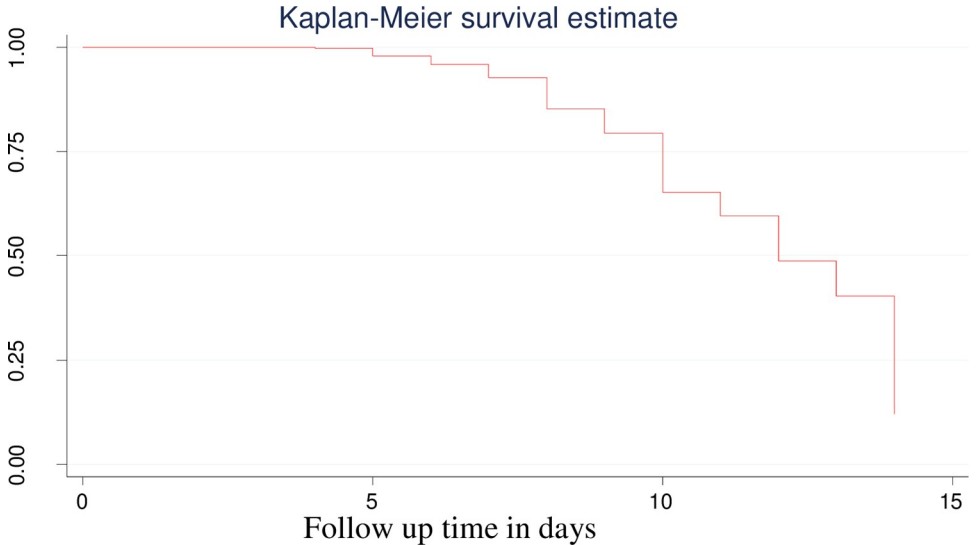

**Fig 2. The overall Kaplan Meier estimate of time to recover from necrotizing enterocolitis admitted from May 2019 to May 2022 at Bahir Dar Public Hospitals.**

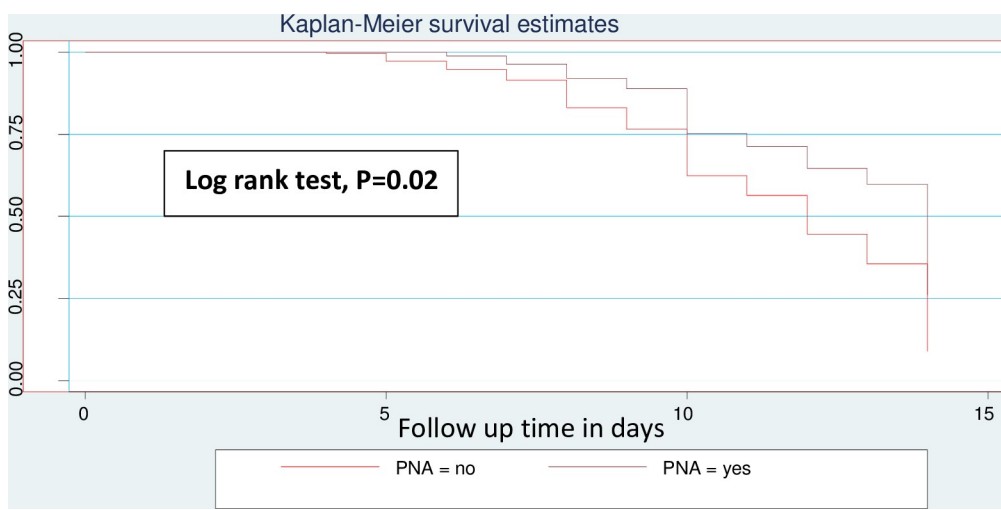

**Fig 3. Kaplan–Meier analysis for the association between time to recovery from NEC and neonates diagnosed for perinatal asphyxia in NICU.**

neonates were died, 10 (2.8%) left against medical advice, and 4 (1.1%) were considered as disappeared.

## Predictors of recovery time from NEC

In a bi-variable Cox regression analysis, determinants such as PNA, weight of the neonate, stage of NEC, type oxygen support, and platelet count of the neonate at admission were associated with time to recovery.

After analysis of multivariable Cox regression, the variables PNA, weight of the neonate, stage of NEC, and platelet count were statistically significant predictors of time to recovery from NEC.

The proportional hazard assumption was satisfied for the significantly associated variables. Neonates with the birth weight of 1500-2499gm were1.94 times (AHR: 1.94, 95% CI: 1.26–2.99) and neonates with the weight of 2500-4200gm were 1.98 times (AHR: 1.98, 95% CI: 1.15–3.42) more likely to recover faster than neonates with a birth weight less than 1500gm.

Neonates with PNA were 49% times (AHR: 0.56, 95% CI: (0.35–0.74)) less likely to recover from NEC compared as neonates who were diagnosed for PNA (Fig 3). Neonates who had platelet count greater than or equals to 150,000 were 1.75 times (AHR: 1.75, 95% CI: 1.24–2.48) more likely to recover faster than neonates who had a platelet count <150,000 (Fig 4). Neonates who were diagnosed as stage III NEC were 0.05% times (AHR: 0.95, 95% CI: (0.69–1.30)) less likely to recover than neonates who were diagnosed as stage I NEC (Fig 5). The predictors of time to recovery from NEC indicated at (Table 5).

## Discussion

The study found that the median time to recover from NEC was 12 days. This helps in setting realistic expectations for recovery duration and feeding initiation timing. The study reported an overall recovery incidence rate of 60 per 1000 person-days, giving insight into the recovery rate in the study population. Notably, the recovery rate of 61.9% after 14 days of follow-up was higher than that reported in a study conducted in Indonesia (44.2%) [19]. The median time for achieving full enteral feeding in the Netherlands was reported as 14 days. In the USA, medical management courses typically last for 10 to 14 days [12, 19]. The recovery time from NEC in

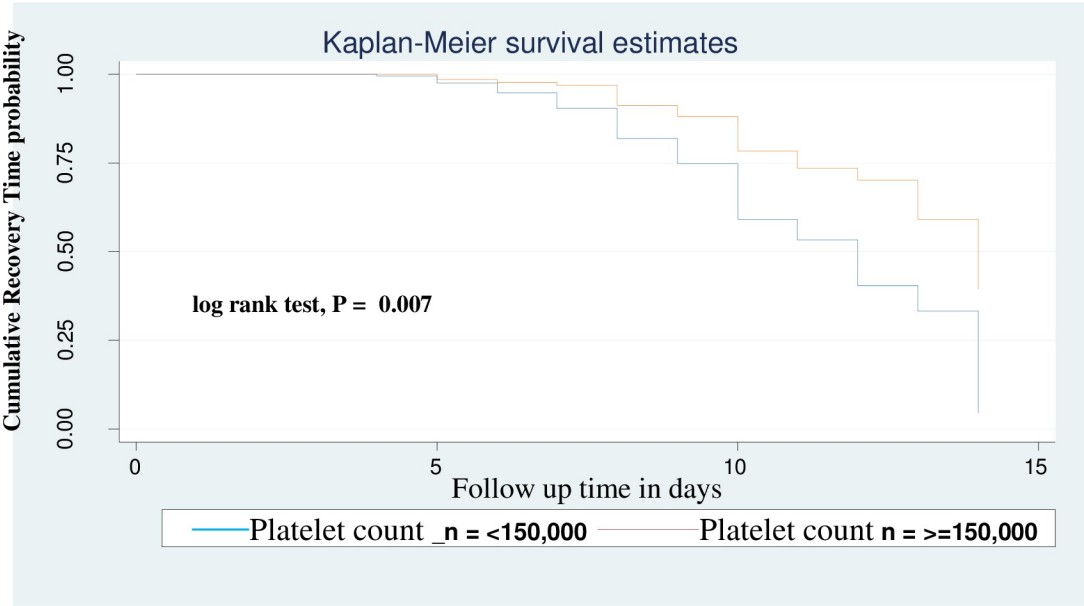

**Fig 4. Kaplan–Meier analysis for the association between time to recovery and platelet count for neonates admitted in NICU.**

this study was longer compared to a previous study conducted in Los Angeles, which reported a timeframe of 5–7 days [35]. This could be due to differences in factors like disease severity, management protocols, access to care, or other social conditions between the study populations [36].

The multivariable analysis identified several significant predictors of time to recovery from NEC. These predictors included platelet count, birth weight, stage of NEC, and comorbidities such PNA. The study's finding showed that platelet count was a significant predictor of recovery time in neonates with NEC. A platelet count greater than 150,000 was associated with

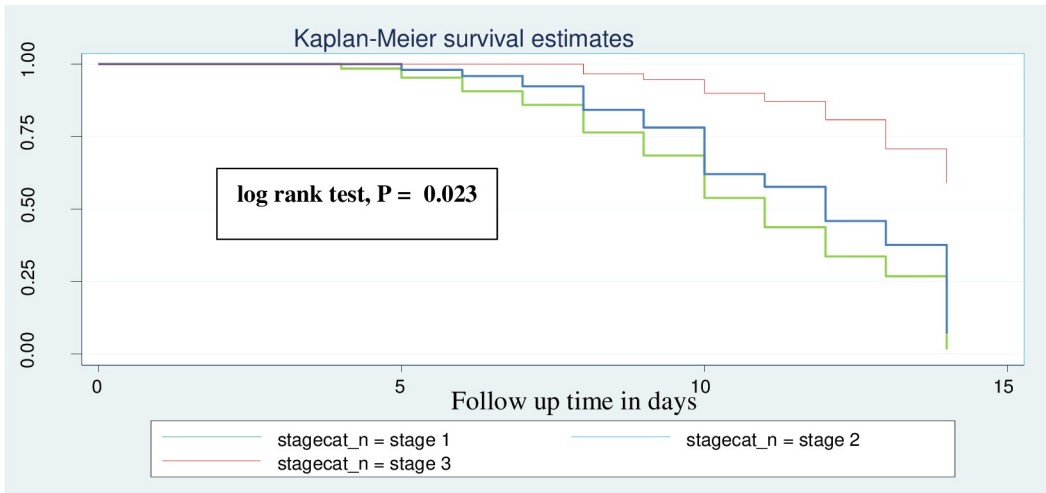

**Fig 5. Kaplan–Meier analysis for the association between time to recovery and necrotizing enterocolitis stage for neonates admitted in NICU.**

**Table 5. Multivariable Cox proportional regression analysis of determinant factors for time to recovery from necrotizing enterocolitis among neonates admitted at Bahir Dar Public Hospitals, Northwest Ethiopia from 2019 to 2022.**

| Variable | | Recovery | Censored | CHR (95%) | AHR (95%) |
|---|---|---|---|---|---|
| HMD | Yes | 142 | 91 | 1 | 1 |
| | No | 81 | 47 | 1.47(1.11,1.93) | 0.86(0.62,1.18) |
| PNA | Yes | 36 | 69 | 0.56 (0.41,0.84)** | 0.51 (0.35,0.74)** |
| | No | 187 | 69 | 1 | 1 |
| Reinitiating day of feeding | <5 | 192 | 109 | 1.54(1.05,2.25) | 1.29(0.86,1.92) |
| | >5 | 31 | 29 | 1 | 1 |
| Platelet count | <150,000 | 41 | 110 | 1 | 1 |
| | >150,000 | 182 | 28 | 2.19(1.56,3.08) | 1.75(1.24,2.48)** |
| Stage of NEC | Stage I | 60 | 5 | 1 | 1 |
| | Stage II | 149 | 58 | 0.76(0.57,1.03) | 0.95(0.69,1.30) |
| | Stage III | 14 | 75 | 0.22(0.12,0.39)*** | 0.42(0.23,0.77)** |
| Weight of neonates | <1500gm | 50 | 99 | 1 | 1 |
| | 1500-2499gm | 90 | 24 | 2.4(1.38,3.36)*** | 1.94(1.26,2.99)** |
| | 2500-4200gm | 83 | 15 | 2.44(1.71,3.46)*** | 1.98(1.15,3.42)* |
| GA | 29-31wk | 29 | 62 | 1 | 1 |
| | 32-37wk | 130 | 52 | 1.79(1.20,2.69) | 0.81(0.48,1.38) |
| | >37 | 64 | 24 | 2.51(1.61,3.89) | 1.2(0.64,2.46) |

*P less than 0.05

**P less than 0.01, and ***P less than 0.001

earlier recovery compared to a platelet count below 150,000. This finding is not consistent with study finding conducted in Indonesia [19]. The discrepancy in these findings may be due to the platelet measurement methodology, the time when the sample is tested after admission, and the quality of the health facility, all of which can influence the recovery time difference for necrotizing enterocolitis [37]. The study identified that birth weight was a significant factor associated with recovery time in neonates with NEC. Neonates with a birth weight in the range of 1500–2499 grams were found to be 1.94 times more likely to recover faster compared to neonates with a birth weight below 1500 grams. The result was consistent with the study done in Indonesia and in China. Weight of the neonate was found to be a significant determinant of the recovery and a longer hospital stay in NEC [19, 38].

In this study the stage of NEC was found to be a significant predictor of recovery time. Neonates with stage I and stage II NEC were recover sooner compared to those with stage III NEC. The more advanced stage of NEC indicates a higher level of disease severity, which can lead to a longer recovery time. The finding that higher stages of NEC are associated with delayed recovery time aligns with the study conducted in Indonesia [19, 39, 40].

In this study, comorbidities like PNA were found to be significant clinical indicators affecting the recovery time from NEC. Neonates with these comorbidities experienced delayed recovery compared to those without them. This finding aligns with a follow-up study conducted in the USA, which also reported delayed recovery in neonates with comorbidities of PNA [20].

## Conclusion

This study found that the median time to recover from NEC was 12 days, with predictors such as neonate weight, platelet count, NEC stage, and comorbidities significantly influencing

recovery time. In light of these results, it is recommended to focus on interventions and strategies aimed at reducing the median time of recovery for NEC, with the goal of improving outcomes and optimizing patient care.

## Limitation of the study

The follow-up time to evaluate recovery from NEC was set at 14 days, based on the literature. The study design was retrospective; we are suggesting the need for a prospective study. To determine the time to reach full eternal feeding it was used, single population mean sample size calculation formula with a margin of error (d = 3), which may not ensure as the calculated sample size quite enough, this was due to luck data to confirm it additionally, due to the lack of literature on the study topic, there was insufficient data available for result comparison in the discussion section. The study used time to full enteral feeding was used to evaluate NEC recovery, but not as the sole or primary indicator, so that we recommend other researcher should be considered in the broader context of the neonates' overall clinical status and other relevant measures of recovery.

## Supporting information

**S1 Data.**
(XLSX)

## Acknowledgments

I would like to thank Debre Birhan University, Aserat Woldeyes Health Science Campus, and Department of Paediatrics and Child Health Nursing for giving us the chance to conduct this study. Also, we would like to express our gratitude for hospitals administrators, staffs and data collectors for their kind contribution in the data collection. We would like to acknowledge the authors of the included articles and Mr. Semahegn G for his English language spelling and grammar edition.

## Author Contributions

**Conceptualization:** Birtukan Ayana Tefera, Abdurahman Mohammed Ahmed, Sisay Shewasinad Yehualashet.

**Data curation:** Birtukan Ayana Tefera, Abdurahman Mohammed Ahmed, Sisay Shewasinad Yehualashet.

**Formal analysis:** Birtukan Ayana Tefera, Sisay Shewasinad Yehualashet.

**Investigation:** Birtukan Ayana Tefera, Sisay Shewasinad Yehualashet.

**Methodology:** Birtukan Ayana Tefera, Abdurahman Mohammed Ahmed, Sisay Shewasinad Yehualashet.

**Project administration:** Birtukan Ayana Tefera, Abdurahman Mohammed Ahmed, Sisay Shewasinad Yehualashet.

**Software:** Abdurahman Mohammed Ahmed, Sisay Shewasinad Yehualashet.

**Supervision:** Sisay Shewasinad Yehualashet.

**Validation:** Birtukan Ayana Tefera, Abdurahman Mohammed Ahmed, Sisay Shewasinad Yehualashet.

**Visualization:** Birtukan Ayana Tefera, Sisay Shewasinad Yehualashet.

**Writing – original draft:** Birtukan Ayana Tefera, Abdurahman Mohammed Ahmed, Sisay Shewasinad Yehualashet.

**Writing – review & editing:** Birtukan Ayana Tefera, Abdurahman Mohammed Ahmed, Sisay Shewasinad Yehualashet.

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
