## [Decision Letter · Decision Letter 0]

16 May 2023

PONE-D-23-07935Time to Recovery from Necrotizing Entrocolites and Its Determinant Factors among Hospitalized Neonates in Bahir Dar, Ethiopia: A Retrospective Follow up Study, 2022.PLOS ONE

Dear Dr. Sisay Shewasinad Yehualashet

Thank you for submitting your manuscript to PLOS ONE. After careful consideration, we feel that it has merit but does not fully meet PLOS ONE’s publication criteria as it currently stands. Therefore, we invite you to submit a revised version of the manuscript that addresses the points raised during the review process.

We look forward to receiving your revised manuscript.

Kind regards,

Sanjoy Kumer Dey, M.D

Academic Editor

PLOS ONE

Journal Requirements:

"no source of funding"

"authors have no competing interests"

6. Please amend either the abstract on the online submission form (via Edit Submission) or the abstract in the manuscript so that they are identical.

Reviewers' comments:

Reviewer's Responses to Questions

**Comments to the Author**

1. Is the manuscript technically sound, and do the data support the conclusions?

Reviewer #1: Yes

Reviewer #2: Partly

2. Has the statistical analysis been performed appropriately and rigorously? 

Reviewer #1: Yes

Reviewer #2: Yes

3. Have the authors made all data underlying the findings in their manuscript fully available?

Reviewer #1: Yes

Reviewer #2: Yes

4. Is the manuscript presented in an intelligible fashion and written in standard English?

Reviewer #1: Yes

Reviewer #2: No

5. Review Comments to the Author

Reviewer #1: Thank you, authors, for bringing up this essential study issue in your scholarly work. The results are valuable for practice and have a bearing on public health. I do have some major and minor concerns, though.

Major concern

1. You determine the recovery of baby from NEC by reaching full calorie intake (150 ml/kg or 180 ml/kg) and limiting the follow-up period to 14 days. The standard duration of therapy for NEC ranges from 7 to 14 days and also in some cases like Surgical NEC, prematurity, Stage III PNA it may take longer than that to achieve complete feeding. Therefore, the majority of the baby may be ignored or recruited to censor, especially in stage II and stage III NEC babies. This could lead to an incorrect conclusion. I am not clear why do you want to restrict the follow-up date?

Minor comments

1. You stated that recovery time is slightly lower in line 43 of your conclusion. What does "slightly lower" mean? And what criteria do you use to determine whether something is lower or higher? It is better to describe in faster or prolonged.

2. If your finding implies faster recovery time, your recommendation may be better relate with enhancing or strengthening the existing practice.

3. Your paper's introduction section is lengthy. Please keep it brief and to the point by combining related results into a single paragraph.

4. You claimed that respiratory assistance was used to help those with respiratory failure in Line 254. You ought to state Respiratory distress, in my opinion. Furthermore, What about resuscitation considering that around 30% of babies have PNA?

5. Line 262 median follow-up time was 10 days. Remove it because it doesn't convey any vital information. Instead, it is enough to describe the incidence density rate and median recovery time.

6. Be sure that the proportional hazard assumption is satisfied, unless using other parametric models is preferable. The PNA KM curve and NEC stage seem to contradict the assumption of hazard proportionality.

7. Make the necessary corrections in Table 4 (Breast Milk), and in Table 6 (Reinitiating Day of Feeding Reference =1), Line 340 (Incidence Rate), Line 345 (Babies or Newborns),

8. Please include some standards or scientific support for your discussion section beyond comparing your findings to those of others. Also, summarize comparable ideas or findings into a single paragraph.

9. Avoid redundancy of ideas. For instance, lines 264 and 285; 271 and table 5; 299 and lines 360–362...

Reviewer #2: Title: Time to Recovery from Necrotizing Enterocolites and Its Predictors among Hospitalized Neonates in Bahir Dar, Ethiopia: A Retrospective Follow up Study.

Abstract:

Line 39-43- All of these predictors have a weak strength of association, do you think, based on such strength your recommendation will be acceptable?

Line 46-49: How can assured that they were not give attention? Your recommendation better to be modified based on your findings, but not based on your real facts.

Introduction:

General comment: It didn’t show the gap that the findings of this research will fill. Previous works of literatures were not dig out well, poor literature review you did. The general write up was not good including the language, better to be modified. The coherence of the write was not good, better to be revised. Finally, better to precisely and coherently written with adequate literature review.

Methods:

Line 160-64: I think such type of sample size determination formula is not appropriate for time to event data. I am eager to hear your justification why you are using margin of error=3???

Line 204: Are they professionals working at the study site?

Line 227: What are the method used for assumption checking for independent variables and when we accept the variable for regression or at least by what method the variable should fulfil the assumption to use cox proportional Hazard regression?

Line 235: Could you think this graph showed the model was fitted? Better to see or clean your data again and again before fitting the model and adequacy checking.

Result:

Line 259: Table 2: In Your statement of justification early age of neonate is risk factors, but here there is less risk difference. How do you justify it? Where did you get the age of the mother, hence it is a chart review for neonates and if there is the chance to get maternal data, why not include maternal related other variables?

Table 3: What is the prevalence of preterm birth in Ethiopia? Have you appreciate your descriptive statistics here? You know that preterm neonates are more prone to develop NEC than others if so is it comparable?

Line 283 and 287: What is the cumulative survival probability? And Did the service provision given at these hospitals reduce the neonatal mortality due to NEC??? Why such large proportion of neonates were died after admission? Have got insights regarding such death?

Line 307: Why most of the neonate were died after admission??? The NEC or the poor service delivery system??? Better to incorporate in your discussion such peculiar findings.

Figure 4: Did the graph fulfil the assumption? Did this graph fulfil the assumption statistically? Better to include statistically methods of assumption checking in addition to graphical methods?

Table 6: PNA is Well Known factor for NEC. Are you sure not having PNA is a cause for NEC??? Needs detail discussion. Most of your statistically significant variables had weak strength of association. How can you interpreted it? Do you think based on this findings recommendation is acceptable?

Discussion:

General comments: your discussion is poor even though your findings are peculiar including your write up, coherence and interpretation and implication. Revised your discussion.

Line 361: If it reported incidence of survival, which was censored, not comparable with your findings since you are reported incidence proportion of recovery, which was event.

Line 372: Do you think sample size difference a justification for result difference? What does sampling distribution mean?

6. PLOS authors have the option to publish the peer review history of their article (what does this mean?). If published, this will include your full peer review and any attached files.

Reviewer #1: No

Reviewer #2: No

---

## [Author Response · Author response to Decision Letter 0]

26 Jul 2023

Dear Academic Editor!

PLOS ONE 

Response to Reviewers

We are pleased to resubmit for publication version of “Time to Recovery from Necrotizing Enterocolites and Its Predictors among Hospitalized Neonates in Bahir Dar, Ethiopia: A Retrospective Follow up Study, 2022.” for a review as original research in PLOS ONE.

The comments of the editor and the reviewers were highly insightful and enabled us to greatly improve the quality of our manuscript. Therefore, based on the editor’s and the reviewers’ concerns we have made extensive edition in our manuscript. Especially we have extensively edited the manuscript by a professional language editor (English-language Teacher (Simegnew) thoroughly edited the manuscript for language usage, spelling, and grammar) before submitting the revised version. We have addressed yours’ concerns in a point by point format. 

 We look forward to hearing from you at your earliest convenience. 

Thank you for your consideration of this manuscript! 

Kind regards,

Sisay S.

On behalf of authors

Editor and Review Comments to the Authors

Reviewer comments and Author response 

Response for the Reviewer #1 comments

Reviewer #1: Thank you, authors, for bringing up this essential study issue in your scholarly work. The results are valuable for practice and have a bearing on public health. I do have some major and minor concerns, though.

Major concern

1. You determine the recovery of baby from NEC by reaching full calorie intake (150 ml/kg or 180 ml/kg) and limiting the follow-up period to 14 days. The standard duration of therapy for NEC ranges from 7 to 14 days and also in some cases like Surgical NEC, prematurity, Stage III PNA it may take longer than that to achieve complete feeding. Therefore, the majority of the baby may be ignored or recruited to censor, especially in stage II and stage III NEC babies. This could lead to an incorrect conclusion. I am not clear why do you want to restrict the follow-up date? Response: Thanks the reviewer for your major concern. As you explained we determined the recovery of babies from NEC by reaching full calorie intake (150 ml/kg or 180 ml/kg). But we were not limited the follow up period in our proposal. But 14 days of follow up period was determined after literature review on the title. Also as you mentioned we agree that on some cases like Surgical NEC, prematurity, Stage III PNA it may take longer than 14 days to achieve complete feeding. Of the 138 censored, 124 (89.9% of the censored) or 124 (34.3%) of the total study participants were died in the follow up period and 10 (7.2%) of centered or 10 (2.8%) of the total cases were resisted/against medical advice. The high mortality rate of neonates with NEC after hospital admission may related to the severity of the disease, quality/standard of care provided in the hospitals. Therefore, this could not lead to an incorrect conclusion. The study's objective was to ascertain whether NEC might recover within 14 days and how long it would take. The median recovery time was 12 days on average (10 to 14 days). From the total of 361 neonate with NEC, 223 (61.8%) were recovered within 14 days. Based on the study finding, fasten recovery will be achieved by treating the predictors, which included Stage of NEC, low birth weight, respiratory distress syndrome, perinatal asphyxia, and decreased level of platelet count. It is crucial to comprehend the length of recovery and predictors in order to prevent nosocomial infection, lower expenses, and better manage resources. We will also indicate this as a limitation (line 391-400) and recommend the researchers use the finding as a basis line data to do prospective cohort studies that extend longer than 14 days.

Minor comments

1) You stated that recovery time is slightly lower in line 43 of your conclusion. What does "slightly lower" mean? And what criteria do you use to determine whether something is lower or higher? It is better to describe in faster or prolonged.

Response: Dear Reviewer, I appreciate your feedback and suggestions. According to your advice, line 43-44 has been changed to median recovery time rather than slower or faster because, according to biostatistics expert Professor Taddese Awoke consent, it is preferable to leave the decision to the scholar in order to avoid making false judgment.

2) If your finding implies faster recovery time, your recommendation may be better related with enhancing or strengthening the existing practice.

Response: the comment accepted and considered in the revised version of this manuscript. It is corrected from line 45-49 and as well at conclusion part. 

3) Your paper's introduction section is lengthy. Please keep it brief and to the point by combining related results into a single paragraph.

Response: as per your comment the introduction section revised by removing redundancy and margining similar idea, so the word count reduced from 2773 to 743 in the revised manuscript (from line 52-96)

4) You claimed that respiratory assistance was used to help those with respiratory failure in Line 254. You ought to state Respiratory distress, in my opinion. Furthermore, What about resuscitation considering that around 30% of babies have PNA?

Response: Thank you for your concern, Neonatal resustation was take place at delivery room, but we consider respiratory support given at NICU with CPAP, INO2 and other means.

5) Line 262 median follow-up time was 10 days. Remove it because it doesn't convey any vital information. Instead, it is enough to describe the incidence density rate and median recovery time.

Response: Thank you, the comment accepted and it is removed, and the corrections take place from line 247-251

6) Be sure that the proportional hazard assumption is satisfied, unless using other parametric models is preferable. The PNA KM curve and NEC stage seem to contradict the assumption of hazard proportionality.

Response: Thank you for your notification, it was also our concern, but as to our consultation with a biostatisticians the PNA KM curve and NEC stage even it seems as it violating the assumption, it is not cross one group to the other comparison group so it satisfied the assumption.

7) Make the necessary corrections in Table 4 (Breast Milk), and in Table 6 (Reinitiating Day of Feeding Reference =1), Line 340 (Incidence Rate), Line 345 (Babies or Newborns),

Response: thank you for your comment and suggestion, all necessary correction was made as per your comment. Breast milk at table 4, Reinitiating Day of Feeding Reference =1 at table 6, incidence rate and newborns was corrected accordingly.

8) Please include some standards or scientific support for your discussion section beyond comparing your findings to those of others. Also, summarize comparable ideas or findings into a single paragraph.

Response: Thank you for your comment, by considered your comment, we reviewed the discussion section and necessary correction was taken from line 303- 384 as per the comment provided.

9) Avoid redundancy of ideas. For instance, lines 264 and 285; 271 and table 5; 299 and lines 360–362...

Response: Thank you, well accepted and correction was made in the revised version of our manuscript. 

Response for Reviewer #2 comments

Dear reviewer thank you, your comment was sound and helpful for the improvement of this manuscript.

Title: 

1) Time to Recovery from Necrotizing Enterocolites and Its Predictors among Hospitalized Neonates in Bahir Dar, Ethiopia: A Retrospective Follow up Study.

Response: thank you for your correction, the modification you made in the title is accepted and we modified on the revised version of our manuscript.

Abstract:

2) Line 39-43- All of these predictors have a weak strength of association, do you think, based on such strength your recommendation will be acceptable?

Response: Even if the correlation is not substantial, all of the factors found here are clinically or biologically feasible, and the 95% confidence intervals for each of them are small, indicating the accuracy of the estimate. In this regard, the recommendations may help to enhance the care given to newborns with NEC. Moreover, we recommended other researchers to conduct study using prospective cohort study design by including other variables not considered in this study. We included this weak association on the limitation of the study line 391-340

3) Line 46-49: How can assured that they were not give attention? Your recommendation better to be modified based on your findings, but not based on your real facts.

Response: The comment accepted and it is modified in the revised version of our manuscript from line 45-49 and on conclusion part. But we recommended like that based on looking to high mortality rate (34.4%) of the cases, which were related with the nature of the disease and severity of the problem, to mean there is a need to strengthening or enhancing the program which can reduce neonatal mortality after admission to hospitals. 

Introduction:

4) General comment: It didn’t show the gap that the findings of this research will fill. Previous works of literatures were not dig out well, poor literature review you did. The general write up was not good including the language, better to be modified. The coherence of the write was not good, better to be revised. Finally, better to precisely and coherently written with adequate literature review.

Response: The comment accepted and the introduction section is modified and we tried to make it precise and coherent from line 52-96

Methods:

5) I think using single population formula, such type of sample size determination formula is not appropriate for time to event data. 

Response: Dear reviewer thank you for your comment and suggestion. When we review different literature they used single population formula calculation by considering survival proportion for events. We communicated with biostatics professional and epidemiologist regarding to single population proportion formula, they said it is possible but it has its own limitation

6) Line 160-64: I think such type of sample size determination formula is not appropriate for time to event data. I am eager to hear your justification why you are using margin of error=3???

Response: Most researchers agree that a margin of error of 3% to 8% at the 95% confidence level is acceptable. The larger the margin of error, the less confidence one should have that a poll result would reflect the result of a census of the entire population. So, the margin of error increases as population variability increases; it decreases as population variability decreases. A smaller margin of error suggests that the survey's results will tend to be close to the correct values. Conversely, larger margin of error indicate that the survey's estimates can be further away from the population values. The margin of error is a measure of reliability. For a given confidence level, the smaller the margin error the more precisely we can pinpoint the true mean. What does a 3% error margin mean? The data was estimated with a 95% confidence level of accuracy within 3 percentage points. When we use such sample size calculation formula. Margin of error of (d=3) is used due to being conventionally agree/ think as it is neutral or not too conservative. Moreover, we used it to increase the sample size because the event occurs rarely. 

7) Line 204: Are they professionals working at the study site?

Response: The data collectors were not professionals working at the study site. For this study experienced data collectors who are working in other sites were requited to minimize bias or systematic errors. 

8) Line 227: What are the methods used for assumption checking for independent variables and when we accept the variable for regression or at least by what method the variable should fulfil the assumption to use cox proportional Hazard regression?

Response: Methods used for assumption checking of independent variables for regression includes in survival analysis, it is highly recommended to look at the Kaplan-Meier curves for all the categorical predictors. This will provide insight into the shape of the survival function for each group and give an idea of whether or not the groups are proportional (i.e. the survival functions are approximately parallel). We also consider the tests of equality across strata to explore whether or not to include the predictor in the final model. For the categorical variables we will use the log-rank test of equality across strata which are a non-parametric test. For the continuous variables we will use a univariate Cox proportional hazard regression which is a semi-parametric model. We will consider including the predictor if the test has a p-value of 0.2 – 0.25 or less. We are using this elimination scheme because all the predictors in the data set are variables that could be relevant to the model. If the predictor has a p-value greater than 0.25 in a univariate analysis it is highly unlikely that it will contribute anything to a model which includes other predictors. It is stated on line 162-197

9) Line 235: Could you think this graph (figure 1) showed the model was fitted? Better to see or clean your data again and again before fitting the model and adequacy checking.

Response: We see that the hazard function follows the 45 degree line very closely except for very large values of time. It is very common for models with censored data to have some twisting at large values of time and it is not something which should cause much concern. Overall we would conclude that the final model fits the data very well, mean that the model was fitted

Result:

10) Line 259: Table 2: In Your statement of justification early age of neonate is risk factors, but here there is less risk difference. How do you justify it? Where did you get the age of the mother, hence it is a chart review for neonates and if there is the chance to get maternal data, why not include maternal related other variables?

Response: As it is stated in the introduction section, NEC usually occurs 2-3 days after birth with 90% of cases occurs during first 10 days of life. As we see from the frequency distribution 258 (71.5%) of the cases occurred in the first 7 days of the neonates (Table 2), for your information 333(92.2%) of the cases of NEC occurred during the first 10 days of life in this study. Concerning the age of the mother we got it from the neonatal chart during the review (registration book, including maternal age). Regarding, why not include maternal related other variables? When we do pre-test for the check list other variables, there were not complete in each neonatal chart, and then we removed from our checklist after the pre-test done. 

11) Table 3: What is the prevalence of preterm birth in Ethiopia? Have you appreciate your descriptive statistics here? You know that preterm neonates are more prone to develop NEC than others if so are it comparable?

Response: Dear reviewer your comment is helpdful, the prevalence of preterm birth in Ethiopia is estimated around 10.48 (95% CI 7.98-12.99) as per the systematic review based on studies done from 2009-2019, which published in 2020 by BMC pregnancy and childbirth. But in this study more than 75% of neonates were less than or equal to 37 weeks. It was high because, this is due to the nature of the disease NEC most commonly affects the preterm neonates. 

12) Line 283 and 287: What is the cumulative survival probability? And Did the service provision given at these hospitals reduce the neonatal mortality due to NEC??? Why such large proportion of neonates were died after admission? Have got insights regarding such death?

Response: cumulative survival probability is the set of probabilities used in estimating the probability of death or survival at each year and the cumulative probability of survival to each year is called a life table. Our way of explanation is completely not correct. So the way of interpretation changed like this “Of 361 neonates were followed the 1st neonate was recovered at the 4th day of the follow-up”. As a researcher we believe that this much death after being admitted at the Hospital’s neonatal intensive care unit is not only due to disease severity but also due to quality of care provision or poor service delivery. That is why 

---

## [Decision Letter · Decision Letter 1]

15 Dec 2023

PONE-D-23-07935R1Time to Recovery from Necrotizing Enterocolites and Its Predictors among Hospitalized Neonates in Bahir Dar, Ethiopia: A Retrospective Follow up Study, 2022PLOS ONE

Dear Dr. Yehualashet,

Thank you for submitting your manuscript to PLOS ONE. After careful consideration, we feel that it has merit but does not fully meet PLOS ONE’s publication criteria as it currently stands. Therefore, we invite you to submit a revised version of the manuscript that addresses the points raised during the review process.

We look forward to receiving your revised manuscript.

Kind regards,

Sanjoy Kumer Dey, M.D

Academic Editor

PLOS ONE

Reviewers' comments:

Reviewer's Responses to Questions

**Comments to the Author**

1. If the authors have adequately addressed your comments raised in a previous round of review and you feel that this manuscript is now acceptable for publication, you may indicate that here to bypass the “Comments to the Author” section, enter your conflict of interest statement in the “Confidential to Editor” section, and submit your "Accept" recommendation.

Reviewer #3: (No Response)

2. Is the manuscript technically sound, and do the data support the conclusions?

Reviewer #3: No

3. Has the statistical analysis been performed appropriately and rigorously? 

Reviewer #3: I Don't Know

4. Have the authors made all data underlying the findings in their manuscript fully available?

Reviewer #3: No

5. Is the manuscript presented in an intelligible fashion and written in standard English?

Reviewer #3: No

6. Review Comments to the Author

Reviewer #3: Time to Recovery from Necrotizing Entrocolites and Its Determinant Factors among Hospitalized Neonates in Bahir Dar, Ethiopia: A Retrospective Follow up Study, 2022

Dear editorial team: Thank you very much for giving me the opportunity to review this manuscript

It is well known that NEC is one of the most common devastating, life-threatening, acquired gastrointestinal disorder in neonates. Despite this fact, why you are interested in studying time to recovery from NEC? Do you think that time is a big deal? Why you failed to deal about treatment outcome or survival status or something else?

From the abstract section: you should show the gaps and why you are going to deal about time to recovery from NEC?

The information presented from the system and the main document is different. So, which abstract is presented for review?

Reference 4 is missed

Background:

Long sentence with poor grammar typographical or grammatical errors should be corrected.

Most paragraphs are constructed with a single sentence and more complicated to understand. I suggest professional help in revising this manuscript. The flow of idea from one paragraph to the other paragraph is unrelated and difficult to understand.

Most of the idea presented in each paragraph is unrelated to the study objectives.

The overall incidence and /or prevalence of NEC is not stated clearly. The gap/why this study was carried out is not also described?

Data extraction period presented from the system and the main document is different. What is the reason for this discrepancy?

Why did you calculate sample size using single population mean rather than double population? In majority of health science studies we use tolerable margin of error (d=5%) but you use 3%. Could you explain your justification? The sample size calculation is still vague.

What is the source/study population?

What is the inclusion and exclusion criterion?

If the data was collected by neonatal nurse, how do you handle different types of bias? Why do you calculate median time instead of mean time to determine survival time?

Abbreviation/acronym should be written in full sentence during the first appearance. Eg FHCSH, TGSTH, AH, HMD, PNA, RDS, and GA

Discussion it needs major revision

First please remove statements described in the first two line of discussion section

When you write discussion, first you should discuss on the recovery time from NEC on similar studies conducted across the globe and put your scientific justification for the possible discrepancy if any then you can discuss on each explanatory variables starting from the local to global.

The statement presented on line number 346 is not similar/comparable with the idea stated on line number 347-49. So, it needs revision.

The statement described from line number 350-353 is not clear and is not presented well.

You are not expected to restate the results of the study rather comparing your study result with the other previously conducted researches and put the evidence for variations.

The possible reason raised during discussion is not sound and persuasive. So, it needs a major modification.

The discussion part is not written adequately and you should write starting from local to global unlike literature review from global to local.

In order to write a good discussions, you should search exhaustively similar studies done a broad and nationally. unless it becomes shallow.

You are also expected to discuss on the implications of the findings in context of existing research.

Based on your findings, you should write appropriate recommendation.

Limitation of the study is not written? Why

7. PLOS authors have the option to publish the peer review history of their article (what does this mean?). If published, this will include your full peer review and any attached files.

Reviewer #3: No

---

## [Author Response · Author response to Decision Letter 1]

4 Jan 2024

Dear Editor and Reviewer,

Thank you for taking the time to review our article. We appreciate your valuable feedback and constructive comments. Your insights have provided us with a deeper understanding of the subject matter and have helped us improve the quality and clarity of our article.

Best regards,

Response for the Reviewer #3 comments

It is well known that NEC is one of the most common devastating, life-threatening, acquired gastrointestinal disorder in neonates. Despite this fact, why you are interested in studying time to recovery from NEC? Do you think that time is a big deal? Why you failed to deal about treatment outcome or survival status or something else?

1. Do you think that time is a big deal? 

Response: As you said NEC is one of the most common devastating, life-threatening, acquired gastrointestinal disorder in neonates. On this study the treatment outcome and survival status of NEC cases were presented in the result section. Also, Knowing the time to event (recovered or death/censed) in resource limited countries have importance to deal with the required resources in treating the NEC cases and giving insight for health care providers and concerned stakeholders on the required time to recover or survive from NEC among neonates. Intestinal recovery after NEC one of the most important predictors influence neonate survival, so timely recovery plays a crucial role in improving outcomes and reducing morbidity and mortality rates. So, one of the primary challenges in NEC management is early detection. Prompt diagnosis allows for timely initiation of treatment, which can significantly impact recovery time. However, the early signs and symptoms of NEC can be nonspecific and easily overlooked, leading to delayed diagnosis and intervention. 

2. Why you failed to deal about treatment outcome or survival status or something else? 

Response: As it is mentioned above the treatment outcome and survival status of NEC cases were presented descriptively. But, predictors of treatment outcomes and survival status were not identified because the primary outcome of the study was to determine the time to recovery from necrotizing entrocolites and its predictor variables. On the current version of our manuscript in the result section on line 268-271 we mentioned that, “Out of 361 neonates, 223 (61.8%) of them were survived, 138 (38.2%) neonates with NEC were censored, from those 124 (34.3%) neonates died and 10 (2.8%) neonates had left against medical advice and 4 (1.1%) was disappeared without physician consultation. Concerning the treatment outcome, each type of treatment options in managing NEC cases with their treatment outcome (being survived or fail to survive) were presented in the result section on line 254-262 and summarized on (Table 4). 

3. From the abstract section: you should show the gaps and why you are going to deal about time to recovery from NEC?

Response: on line 21-23 the following sentence included, the time it takes for neonates to recover from NEC can vary widely, depending on the severity of the disease, the promptness of diagnosis, and the effectiveness of treatment interventions

4. The information presented from the system and the main document is different. So, which abstract is presented for review?

Response: Thank you, I will submitted the same document on the system and the manuscript 

5. Reference 4 is missed

Response: Reference 4 was cited on line 56, in the introduction section, for the first time. 

Background:

Long sentence with poor grammar typographical or grammatical errors should be corrected. Most paragraphs are constructed with a single sentence and more complicated to understand. I suggest professional help in revising this manuscript. The flow of idea from one paragraph to the other paragraph is unrelated and difficult to understand.

6. Most of the idea presented in each paragraph is unrelated to the study objectives.

Response: thank you very much for your important comment. We acknowledge that English is not our first language and we have edited the manuscript by an English language instructor Ass/professor Simegnew (Debre Berhan University staffs). Based on your recommendations, we have made revision of the manuscript with yellow color accordingly. 

7. The overall incidence and /or prevalence of NEC are not stated clearly. The gap/why this study was carried out is not also described?

Response: thank you very much for your constructive comment that strengthens our manuscript: we incorporated prevalence and incidence of NEC data from line 91 -106. 

8. Data extraction period presented from the system and the main document is different. What is the reason for this discrepancy? 

Response: Thank you, we will submitted the same document on the system and the manuscript

9. Why did you calculate sample size using single population mean rather than double population? In majority of health science studies we use tolerable margin of error (d=5%) but you use 3%. Could you explain your justification? The sample size calculation is still vague.

Response: Dear respected reviewer, it is known that time to event data/study has specific sample size determination formula. But we can’t to get the required information to apply that formula during proposal preparation and assuming as time is a mean type of data we use single population mean formula to determine the sample size. Concerning using a tolerable margin of error (d=3), as you know it is possible to use a margin of error 1-5 for mean type of data or 1% - 5% for proportion type of data. But we used tolerable margin of error (d=3) due to being conventionally agree/ think as it is neutral or not too conservative. Moreover, we used it to increase the sample size and enhance the power of the study. 

Furthermore, you are right, in majority of health science studies we use tolerable margin of error (d=5%) for proportion type of data. But as to my knowledge in advanced biostatistics course use of tolerable margin of error (d=5%) is correct and acceptable if the proportion of the study variable ranges from 0.4 to 0.6 (40% to 60%). Also, it is recommended to use tolerable margin of error (d=1%) if the proportion is less than 0.1 (10%) or proportion greater than 0.9 (90%), which enables as to get enough number of case or events. 

10. What is the source/study population?

Response: Source or target population is the total population interest that have something in common for which the information is required and we wish to draw conclusions/ generalizations at a particular time. Study population is the subset of the source/ target population from which a sample will be drawn or the specific population from which data are collected. 

In our study, “the source population was all neonates with necrotizing enterocolites who were admitted to neonatal intensive care unit of Hospitals in the study area. The study population was all neonates with necrotizing enterocolites who were admitted to neonatal intensive care unit of the study Hospitals during the study period, which was from May 2019 to May 2022.” This description of source and study population included on line number 133-137 in the methods section. 

11. What is the inclusion and exclusion criterion?

Response: Inclusion and exclusion criterion are conditions which indicates the eligibility criteria, which determine the members of the target population who can or can’t participate in the study. 

In our study, “All neonates with necrotizing entrocolites who were treated in neonatal intensive care unit of Felege Hiwot comprehensive specialized Hospital (FHCSH), Tibebe Gion Specialized Teaching Hospital (TGSTH) and Addisalem Hospital (AH) in period from May 2019 to May 2022 were included in the study. But all neonates with necrotizing entrocolites who were treated in neonatal intensive care unit of the three Hospitals and who had incomplete medical record or charts for major variables (date of diagnosis, date of reaching full enteral feeding) were excluded from the study.” This description of included criteria in the current version of our manuscript on line 137-139 in the methods section. 

12. If the data was collected by neonatal nurse, how do you handle different types of bias? Why do you calculate median time instead of mean time to determine survival time?

Response: We assumed that neonatal nurses were close to understand subject matter and had better practical experience on different medical chart which used at NICU, which enter enhance quality of the data. The data collectors were not professionals working at the study site. For this study experienced data collectors who are working in other sites were requited to minimize bias or systematic errors. Also, they are trained and the investigators undertaken re-check on randomly selected the filled data extraction form for its precision. 

13. Abbreviation/acronym should be written in full sentence during the first appearance. Eg FHCSH, TGSTH, AH, PNA, RDS, and GA

Response: Thank you for your comment, I try to correct accordingly on the manuscript part and highlighted with yellow colour: Felege Hiwot comprehensive specialized hospital (FHCSH), Tibebe Gion specialized Teaching hospital (TGSTH) , Addisalem hospital (AH), GA (Gestational age), RDS (Respiratory Distress Syndrome), PNA(Perinatal Asphyxia), MAS(Meconium Aspiration syndrome)

Discussion it needs major revision

14. First please remove statements described in the first two line of discussion section

Response: Thank you for your comment, the first two lines is removed. 

We know that, it is important to include clear and comprehensive information in the discussion section in order to facilitate comparisons with other findings, but it is true that the absence of specific studies on recovery time can make the comparison challenging, We used the following strategies on the discussion part: We acknowledging that the lack of studies on recovery time on the limitation part to comparison this study with other findings. This demonstrates transparency and awareness of the research gap. We try to discuss the finding contextually thorough description of the methodologies, variables, and outcomes used in our study, so recovery time is not explicitly addressed. We discuss finding with related studies that are tangentially related to recovery time or explore similar concepts. This enables us to draw comparisons indirectly and highlight any parallels or discrepancies with the results. In the discussion part we consider broader implication of the finding beyond recovery time. We try to explore how the research finding contributes to the overall understanding of the topic, potentially shedding light on related factors that influence recovery or providing insights into other aspects of the condition or intervention being studied. On the conclusion part we suggest future research direction that could specifically investigate recovery time or address other gaps in the literature and connecting our finding with the existing body of knowledge, even if direct comparisons with other findings on recovery time are not possible.

15. The statement presented on line number 346 is not similar/comparable with the idea stated on line number 347-49. So, it needs revision. 

Response: Thank you it is corrected from line 362-364

16. The statement described from line number 350-353 is not clear and is not presented well.

Response: Thank you it is corrected from line 371-379

17. You are not expected to restate the results of the study rather comparing your study result with the other previously conducted researches and put the evidence for variations.

Response: thank you for your constructive comments: In this particular context, the absence of existing literature or research on the subject makes it challenging to draw upon established findings or expert opinions. So, we try to make our discussion from related or tangential information to formulate a justifiable perspective. And also we consider alternative sources of information, including anecdotal evidence, expert opinions, empirical observations, or analogous studies from related fields. We stated the gab on the limitations part of study. Although the absence of literature poses challenges, it should not discourage us from presenting a well-reasoned justification. By applying logical thinking, exploring alternative sources of information, and recognizing the need for further research, we can still provide a thoughtful and valid perspective on the subject matter at hand.

18. The possible reason raised during discussion is not sound and persuasive. So, it needs a major modification.

Response: Based on your comment we made major modification on the discussion part from line 334-425

19. The discussion part is not written adequately and you should write starting from local to global unlike literature review from global to local.

Response: Thank you for your guidance and it is corrected accordingly 

20. In order to write a good discussions, you should search exhaustively similar studies done a broad and nationally. Unless it becomes shallow.

Response: We conducted an extensive search of open-access journals and databases but have been unable to find a similar study. However, it's important to note that the absence of similar studies doesn't necessarily mean that our research is invalid but we used different discussion writing mechanism 

21. You are also expected to discuss on the implications of the findings in context of existing research.

Response: We incorporated the finding implication to practice and context of existing research, on the discussion part from line 334-425

22. Based on your findings, you should write appropriate recommendation. 

Response: Thank you; we incorporate recommendation from line 456-488

23. Limitation of the study is not written? Why

Response: Response: Thank you; we already stated limitation from line 489-499

---

## [Decision Letter · Decision Letter 2]

26 Mar 2024

PONE-D-23-07935R2Time to Recovery from Necrotizing Enterocolites and Its Predictors among Hospitalized Neonates in Bahir Dar, Ethiopia: A Retrospective Follow up Study, 2022PLOS ONE

Dear Dr. Yehualashet,

Thank you for submitting your manuscript to PLOS ONE. After careful consideration, we feel that it has merit but does not fully meet PLOS ONE’s publication criteria as it currently stands. Therefore, we invite you to submit a revised version of the manuscript that addresses the points raised during the review process.

We look forward to receiving your revised manuscript.

Kind regards,

Sanjoy Kumer Dey, M.D

Academic Editor

PLOS ONE

Reviewers' comments:

Reviewer's Responses to Questions

**Comments to the Author**

1. If the authors have adequately addressed your comments raised in a previous round of review and you feel that this manuscript is now acceptable for publication, you may indicate that here to bypass the “Comments to the Author” section, enter your conflict of interest statement in the “Confidential to Editor” section, and submit your "Accept" recommendation.

Reviewer #4: (No Response)

2. Is the manuscript technically sound, and do the data support the conclusions?

Reviewer #4: Partly

3. Has the statistical analysis been performed appropriately and rigorously? 

Reviewer #4: No

4. Have the authors made all data underlying the findings in their manuscript fully available?

Reviewer #4: Yes

5. Is the manuscript presented in an intelligible fashion and written in standard English?

Reviewer #4: No

6. Review Comments to the Author

Reviewer #4: Specific and detailed comments to the authors were attached as a word file to the system. Thus, I don't think that it's appropriate to repeat it here.

7. PLOS authors have the option to publish the peer review history of their article (what does this mean?). If published, this will include your full peer review and any attached files.

Reviewer #4: **Yes: **Migbar Sibhat

---

## [Author Response · Author response to Decision Letter 2]

8 May 2024

I am writing in response to the comments provided by the reviewer regarding our manuscript titled Time to Recovery from Necrotizing Enterocolitis and Its Predictors among Neonates Admitted to Neonatal Intensive Care Unit in Bahir Dar, Ethiopia: A Retrospective Follow up Study, 2022 (PONE-D-23-07935R2).

We appreciate the time and effort the reviewer has dedicated to reviewing our work, and we would like to thank you and the reviewer for the opportunity to revise and resubmit our manuscript. We have prepared a point-to-point response addressing all the comments and suggestions provided by the reviewer. The manuscript outlines the changes made in the revised manuscript, along with detailed explanations for each modification

General Comment: 

1. “enterocolites” to “enterocolitis” (revise this thourghout the whole manuscript file). 

Response: Thank you for your valuable feedback. I have reviewed the comment and made the necessary revisions throughout the entire manuscript file. The term "enterocolites" has been corrected to "enterocolitis" wherever it appeared. Please let me know.

2. The manuscript had numerous typographic errors (punctuation, grammar, and spelling errors). There are alot of subject-verb agreement errors, numerous tense problems, fragment sentences, active-to-passive voice shifts, and punctuation errors and so on. Hence, I advice the authors to use valid language editor tools or at least invite native language experts. Otherwise, it’s difficult to address each and every edition and typographic errors here as a reviewer. The whole document needs to be exhaustively revised cautiously.

Response: We appreciate the reviewer's feedback regarding the English grammar edition of the manuscript. We understand the importance of ensuring the accuracy and clarity of the language used in the document. We apologize for any typographic errors, punctuation, grammar, and spelling mistakes that may have been present. To address these concerns, we enlisted the assistance of a qualified English instructor, Semehagn.G, from Debre Berhan University, who provided language editing services for the manuscript. However, we acknowledge that there may still be areas that require further attention. In light of the reviewer's recommendation, we will take additional measures to ensure the manuscript undergoes a thorough and cautious revision. We were consider using valid language editor tools and we were make every effort to address all the necessary editions and typographic errors to improve the manuscript's clarity and readability.

3. The manuscript is too long. Please try to reduce several contents and shorten it as much as possible.

Response: Thank you, as per your suggestion we tried to review and shorten the manuscript. By eliminating redundancies, reorganizing sections, and focusing on the most essential points, we make the manuscript more concise. 

Topic: 

4. I suggest to replace the word “hospitalized” with the term “admitted to the neonatal ICU” in Bahir Dar. Hospitalized is a vague term and doesn’t specifically represent neonates admitted to the ICU but all who visited the hospital for any reason with or without admission. 

Response: Thank you very much, the topic modified as per the suggestion like this: “Time to Recovery from Necrotizing Enterocolitis and Its Predictors among Neonates Admitted to Neonatal Intensive Care Unit in Bahir Dar, Ethiopia: A Retrospective Follow up Study, 2022”

In the Abstract part

5. Line 1-9: The gap statement (why it is needed to investigate the time to recovery from NEC) should be clearly stated in the background sub-heading of the abstract part. The authors focus on dealing with NEC itself rather than the timing of recovery. 

Response: Thank you, it is well accepted, on line 19-22 the following sentence included, the time it takes for neonates to recover from NEC can vary widely, depending on the severity of the disease, the promptness of diagnosis, and the effectiveness of treatment given for the patients.

6. In the results sub-heading of the Abstract section, the identified predictors are not consistent. For example, RDS & PNA are expected to have negative effect and Platelet count and Stage I NEC are expected to predict recovery positively. Please try to make your findings and their interpretation consistent. Either use positive effects or negative effects by adjusting the reference category while running multivariable Cox-PH regression.

Response: Thank you very much, the comment accepted and the reference category changed for PNA and stage of NEC. And the direction of the association indicated. As you explained, occurrence of PNA and stage III NEC status had negative effect on the time to recover from NEC. But platelet count greater than 150,000 and birth weight other than very low birth weight had positive effect on the time to recover from NEC. This is mentioned in the abstract and also the result section of the manuscript. 

7. Again in the conclusion section of Abstract, consistency rule is violated yet (e.g. NEC stage 1 vs advanced NEC, lower Plt count vs Plt>150,000...). You presented the findings positively in the result part but summaries were drawn negatively. Be consistent in the interpretation of your findings!

 Response: Well accepted and modified as per the comment. We have revised the conclusions to accurately reflect the positive or negative aspects of the results consistently.

8. “The median time to recovery from NEC was 12 days.” This statement could not be a summary/conclusion. It’s just a reptition of your findings. Conclusions should be interpretations of the findings rather. Please revise accordingly.

 Response: Well accepted and modified as per the comment on line 35-39. 

9. “The ministry of health and other stakeholders should improve standard care practice...” Which standard care practice are you going to recommend to be improved? Please try to make it specific and in line with your findings during recommendation. None of your findings assessed and reported ‘standared care practice issues’; and thus, recommending this issue is not part of your objective.

Response: Well accepted and modified/ the recommendation removed as per the comment. We appreciate your comment and agree that recommendations should be based on the findings and objectives of the study.

In the Introduction section:

10. The backround section focuses on describing the fatality and time to death of NEC as well as its devastating consequences on healthcare outcome and costs as well. This was nice. Thus, after reading the study background, readers expect to see in the same way from your study results. However, your ‘Title’ and your study findings contrarily focused on ‘time to recovery’ rather than ‘time to death and fatality’ (Lines 60-82). I suggest re-write either of the two parts (either the background or amend your topic and findings from results through discussion and conclusion). 

Response: Thank you for your valuable feedback regarding the alignment between the background section and the title of our study. We apologize for any confusion caused by the discrepancy between the background section and the title of our study. We are stating the absence of similar studies on the topic, making it challenging to address the concerns rose regarding the background section on the limitation part, but based on your comment, we have made efforts to amend the background section, from line 40-81. 

11. The authors focused on describing the burden of NEC exhaustively. Nevertheless, they failed to state/address the main problem/focus of the study, which was time to NEC rather than the magnitude or incidence of NEC. Thus, the whole introduction section requires thourough revision in this context. 

Response: Thank you for your comment. We appreciate your insightful comments regarding the focus and clarity of the introduction section. We apologize for the oversight in not clearly stating and addressing the main problem of the study, which is the time to necrotizing enterocolitis (NEC) rather than the magnitude or incidence of NEC. Based on your feedback, we were thoroughly revising the introduction section to ensure that it clearly highlights the main problem and focus of the study. 

12. Lines 91-105: These statements can be merged with the 2nd paragraph of the introduction part since the incidence and magnitude are ways of describing diseases burden (morbidity and mortality). Avoid redundancies please!

Response: Thank you for your comment. We appreciate your suggestion to merge the statements in lines 91-105 with the second paragraph of the introduction to avoid redundancies. Based on your suggestion, we were carefully review the content in lines 91-105 and the second paragraph of the introduction. We removed redundancies, by merging these statements. We will make the necessary revisions to enhance the clarity and flow of the introduction section.

13. Line 109-118: “Even with early aggressive treatment, NEC still progresses and has significant morbidity with chronic gastrointestinal complications and poor neurodevelopment....”. Does it mean those adverse health outcomes related to the time of NEC recovery or the disease (NEC) itself? Rather you better state your gap statement specific to your topic of interest (time to NEC recovery) rather than NEC in general. For example, interms of longer recovery vs shorter recovery time after the diagnosis of NEC. Otherwise, your backround and statement will be suitable to study incidence of death/recovery from NEC itself rather than the timing (time of recovery). The two concepts are quite different! Please try to modify accordingly.

Response: Thank you for your comment and we deleted the paragraph.

In the Methods and materials section

14. Lines 121-22: “Institution-based retrospective follow-up study was conducted among NEC neonates in the early neonatal period...”. What does it mean early neonatal period? Does it mean you didn’t consider cases in the late neonatal period? If so you need to revise your population below and results as well. Anyways, be notified and make the required changes accordingly. 

Response: Well accepted and all neonates who were admitted in NICU included in this study. Therefore the word early neonates deleted from the document and edited from line 84-85. 

15. Line 125: “An institution-based retrospective follow-up study was conducted…” avoid redundancy. Please consider deletion. 

Response: well accepted and it is deleted. 

16. Line 140-144: I have a major concern on the appropriateness of the formula that you have used to calculate the required sample size. First, since your study is a follow-up study, single population proportion formula cannot handle/consider follow-up groups equally in the sample estimation process, and thus, is not suitable for such type of studies. Conversely, there are appropriate and scientifically sound methods of sample size determination for survival studies including Cox-model and log-rank methods among others. Why don’t you use one of these methods? Secondly, you couldn’t use a variable that you didn’t assess for sample size estimation. In your case, you used “time to reach a full enteral feeding” for sample size calculation, and you miss it in your results part though. Please try to forward if you have convincing and statistically acceptable rationale for those points!

Response: Dear respected reviewer, as you mentioned it is known that time to event data/study has specific sample size determination formula. But we can’t to get the required information to apply that formula during proposal preparation due to scarcity od directly related articles and assuming as time is a mean type of data we use single population mean formula to determine the sample size with using a tolerable margin of error (d=3), as you know it is possible to use a margin of error 1-5 for mean type of data. But we used tolerable margin of error (d=3) due to being conventionally agree/ think as it is neutral or not too conservative. Moreover, we used it to increase the sample size and enhance the power of the study. Related with your second concern using the variable “time to reach a full enteral feeding” for sample size calculation is due to that, the “time to reach a full enteral feeding” by definition is to mean being recovered for NEC, which is our outcome variable. So, “time to reach a full enteral feeding” was assessed in our study. It is not missed. 

17. Line 157-59: “Necrotizing enterocolitis was defined as an acute disorder of the gastrointestinal in the neonatal period who presented with clinical signs or radiographic findings of NEC and who diagnosed by attending physician.” This definition is pure scientific definition and could not be considered as operational definition/measurement. Furthermore, how did you exactly diagnose NEC for the purpose of your study (Radiology findings, blood tests, clinically or what? What were the criteria used to declare whether the newborn had NEC or not? This should be clearly specified. Because sometimes cases which were diagnosed as NEC by the clinician/physicians in the charts might not fulfill the criteria of NEC diagnosis for your study (false diagnosis) whereas some others might fulfill the criteria despite the clinicians missed it actually (unrecognized cases). Therefore, please revise this section as required, and otherwise it will still be a major concern that could endanger the validity of your findings! Finally, cite the appropriate reference for the definitions provided. 

Response: Thank you for raising important concerns regarding the definition and diagnosis of necrotizing enterocolitis (NEC) in our study. We acknowledge the importance of clearly specifying the criteria used for diagnosis and providing appropriate references. We apologize for any ambiguity in our previous description.

In our study, we defined NEC as an acute disorder of the gastrointestinal tract in the neonatal period, characterized by clinical signs or radiographic findings consistent with NEC, and diagnosed by the attending physician. We understand that this definition may not provide the level of operational detail required for research purposes.

To address your concerns and ensure the validity of our findings, we will provide a more comprehensive and specific description of the diagnostic criteria used for NEC in our study. This will include information on the specific diagnostic methods employed, such as radiology findings, blood tests, clinical assessment, or any other relevant criteria. We will also include appropriate references to support our definition and diagnostic approach.

We appreciate your valuable input, and your comments will help us improve the clarity and validity of our study. We will make the necessary revisions to address the concerns you have raised regarding the definition and diagnosis of NEC.

18. Line 173-76: “Apgar (Activity, Pulse, Grimace, Appearance and Respiration) is a quick assessment performed on a baby at 1, 5 and 10th minutes after birth and it has three levels of score: Low APGAR score 0-3 - Moderate APGAR score 4-6 - Normal APGAR score 7-10.” Please cite the source. 

Response: Thank you for your comment and the reference is cited on line 127-130. 

19. Line 197-98: “Kaplan Meier failure curve was used to estimate the time to 198 recovery during the follow-up.” Why you prefer to use the failure cureve? Time to recovery should be best expressed in the survival function rather than failure. So why you preferred the failure curve?

Response: Kaplan-Meier curve is actually a survival curve that estimates the probability of an event (such as recovery) occurring over time. It is commonly used to analyze time-to-event data, including time to recovery. The term "failure" in this context refers to the occurrence of the event (i.e., recovery) rather than a negative outcome. The Kaplan-Meier curve is preferred because it provides a comprehensive visualization of the probability of recovery over time for the study population. We were editing based on your comment. 

20. Lines 209-210: “The Cox proportional hazard model assumption was checked for variables in the Log-log plot (graphically),...”. Log-log plot is not an assumption test, it is a model fitness check rather. Revise it please.

Response: Dear 

---

## [Decision Letter · Decision Letter 3]

26 Jun 2024

PONE-D-23-07935R3Time to Recovery from Necrotizing Enterocolitis and Its Predictors among Neonates Admitted to Neonatal Intensive Care Unitin Bahir Dar, Ethiopia: A Retrospective Follow up Study, 2022PLOS ONE

Dear Dr. Yehualashet,

Thank you for submitting your manuscript to PLOS ONE. After careful consideration, we feel that it has merit but does not fully meet PLOS ONE’s publication criteria as it currently stands. Therefore, we invite you to submit a revised version of the manuscript that addresses the points raised during the review process.

We look forward to receiving your revised manuscript.

Kind regards,

Sanjoy Kumer Dey, M.D

Academic Editor

PLOS ONE

Reviewers' comments:

Reviewer's Responses to Questions

**Comments to the Author**

1. If the authors have adequately addressed your comments raised in a previous round of review and you feel that this manuscript is now acceptable for publication, you may indicate that here to bypass the “Comments to the Author” section, enter your conflict of interest statement in the “Confidential to Editor” section, and submit your "Accept" recommendation.

Reviewer #4: (No Response)

2. Is the manuscript technically sound, and do the data support the conclusions?

Reviewer #4: Partly

3. Has the statistical analysis been performed appropriately and rigorously? 

Reviewer #4: No

4. Have the authors made all data underlying the findings in their manuscript fully available?

Reviewer #4: Yes

5. Is the manuscript presented in an intelligible fashion and written in standard English?

Reviewer #4: No

6. Review Comments to the Author

Reviewer #4: Title: Time to Recovery from Necrotizing Enterocolitis and Its Predictors among Neonates Admitted to Neonatal Intensive Care Unitin Bahir Dar, Ethiopia: A Retrospective Follow up Study, 2022 (R2)

I would like to thank the editor for offering this opportunity to review the revised version of this work. The authors tried to partially address the comments raised in the previous review session. I appreciate the efforts devoted so far. However, still there are fundamental issues that are not yet moderated, or some unsatisfactory responses forwarded. Without addressing those issues exhaustively, the validity of the study findings will be threatened and not worth for publication.

Major comments:

1.

1.1. The sample size issue remains the major concern in this manuscript. The authors could not forward statistically convincing and satisfactory justification to the previously raised concerns. It is true that single population formula can be applied if there are not studies conducted to fulfill the parameters required to apply the standard methods of sample size calculation for the given study type. However, since the authors were working on dichotomous outcome variable and categorical data, it’s completely unacceptable to use population mean. Population proportion should be the only option in such scenarios if it’s mandatory to use single population formula. How could you apply linear assumptions and parameters to estimate for binary outputs? I doubt the reproducibility of the finding since the applied sample size is one of the major factors that influence the generalizability/representativeness of the study findings.

1.2. The other thing is that “time to reach full enteral feeding” might not necessarily imply NEC recovery since NEC is not the orphan determinant of enteral feeding. I.e neonates without NEC might also encounter such problems. Inability to full enteral feeding could not be generalized to NEC since it could also be due to other causes.

2. The other major concern was that the authors could not provide clear diagnostic criteria applied to identify NEC in their study. The scientific definition provided was vague and non-specific. The diagnosis set by physicians for treatment purpose could not be directly applied for the research purpose. Rather the authors should set specific and clear criteria and differentiate NEC cases during data extraction based on the predetermined criteria (operational definition) from the charts (patient records). You declared the newborn when you got which radiorgraphic findings, which symptoms and signs? as well as how many of those criteria fulfilled?

3. Figure 1 (KM-curve): Wrong graph label and interpretation!

3.1. The x-axis label (recovery time from NEC) is incorrect and that shoud be replaced with “analysis time”. Because the analysis time incorporates the time at risk for both recovered and censored cases (hence could not be exclusively considered as recovery time).

3.2. Line 219-21: “The overall Kaplan Meire estimates identified that the probability recovery of neonates admitted with NEC was zero in the first 3 days of follow up…”. This statement is completely wrong. Any individual who understand English language could not interpret it as the authors intend to infer, NEVER! The authors made incorrect interpretation of the KM-graph. This is because the authors understand the statistical science, but they miss the nature of the event of interest in their own study. Researchers and statisticians need to interpret findings based on the nature of the outcome/event of interest being investigated.

Correct interpretation of the finding in the Figure 1: As we note from the figure (KM-curve), the survival probability (probability of recovery) remains 1 (100%) in the first 3 days, after which it slowly dropped down stepwise as the survival time increased from 0 to 14 days. This means in the first three days; every neonate has the probability to recover from NEC.

4.

4.1. Most of the rationales stated in the discussion section of this manuscript were vague and non-specific to NEC. Majority of the explanations for the discrepancies were general views such as socio-economic, study design, setting difference, infrastructure, quality of healthcare services and so on. However, all these concepts were not declared to have direct association with NEC and the authors failed to cite the sources of such arguments if available.

4.2. Citation of references for the scientific justifications is required yet and it could not be bypassed at all. As you already stated, the purpose of the discussion section is to interpret the results and provide an analysis of the findings in the context of existing knowledge and theories. Please cite those existing knowledge and theories that could support your justification!

Note: Research discussion should address four major points: result interpretation, comparison, justification of discrepancies and explanation of associations, as well as implication to clinical practice. The discussion should enrich with tangible evidences and explanations so that it can be trustworthy to the scientific community. However, if it is solely written based on authors’ opinion and thoughts without tangible and citable sources, it could not be considered as scientific paper. It could be a fiction rather!

Minor comments:

1. Line 69-73: Please remove the first two statements of from the last paragraph of the “Introduction” section. I wonder why the authors just incorporate those statements, while it’s known that NEC is a different concept to asphyxia and RDS.

2. The authors tried to moderate the gap statement. Nevertheless, the manuscript will be benefited if the authors incorporate data related to the difference in survival rate, risk/occurrence of complications and other adverse events between longer vs shorter recovery time. For instance, neonates who recovered fast from NEC might have good prognostic outcome in terms of cognitive function, growth and development, and other long term consequences (NB: This is just an instance).

3. Line 157-58 & Line 162-63: Contradicting ideas presented yet again. Schoenfield Residuals test is an assumption (PH-assumption) test and you stated in the earlier statement as “an assumption test” and later as a “model fitness” test. Please avoid such statistical ambiguities.

4. Line 163-65: Where is the Cox-Snell residual graph? Please provide the graph and cite in “Line 165” as figure file (e.g. Fig. 1).

7. PLOS authors have the option to publish the peer review history of their article (what does this mean?). If published, this will include your full peer review and any attached files.

Reviewer #4: **Yes: **Migbar Sibhat

---

## [Author Response · Author response to Decision Letter 3]

24 Jul 2024

Title: Time to Recovery from Necrotizing Enterocolitis and Its Predictors among Neonates Admitted to Neonatal Intensive Care Unit in Bahir Dar, Ethiopia: A Retrospective Follow up Study, 2022 (R2)

Dear Reviewer, I would like to thank the editor for offering this opportunity to review the revised version of this work. The authors tried to partially address the comments raised in the previous review session. I appreciate the efforts devoted so far. However, still there are fundamental issues that are not yet moderated, or some unsatisfactory responses forwarded. Without addressing those issues exhaustively, the validity of the study findings will be threatened and not worth for publication. 

Major comments:

1. 

1.1. The sample size issue remains the major concern in this manuscript. The authors could not forward statistically convincing and satisfactory justification to the previously raised concerns. It is true that single population formula can be applied if there are not studies conducted to fulfill the parameters required to apply the standard methods of sample size calculation for the given study type. However, since the authors were working on dichotomous outcome variable and categorical data, it’s completely unacceptable to use population mean. Population proportion should be the only option in such scenarios if it’s mandatory to use single population formula. How could you apply linear assumptions and parameters to estimate for binary outputs? I doubt the reproducibility of the finding since the applied sample size is one of the major factors that influence the generalizability/representativeness of the study findings. 

Response: Dear respected reviewer we recognize your great concern on sample size determination formula we used. We used single population mean sample size determination formula specifically to determine the time it takes the neonates to reach full enteral feeding/ the time to recover, because it is one of our specific objectives. Our first objective of the study was not to determin recovery status (yes or no) and that is why we did not used single population proportion formula to estimate the required sample size. But, as you clearly mentioned above the single population mean or even single population proportion sample size calculation formula doesn’t ensure as the calculated sample size is adequate enough for identifying predictors’ variables (our second objective, which was to identify predictors of time to reach full enteral feeding/ recovery), on which the outcome variable is dichotomized (recovered/event and not recovered/censored). Therefore, we decide to mention as a limitation as we did not determine the sample size for our second objective due scarcity of the information. It is mentioned in this version of our manuscript 

Dear respected reviewer, concerning your doubt on the reproducibility and the generalizability/ representatives of the study finding in relation to adequacy of the sample size, as you know even we did not calculated sample separately for our second objective you can see how the 95% confidence intervals are wide or narrow to evaluate the preciseness/ reproducibility/repeatability of the estimate/s. so, as per this study finding all calculated 95% confidence intervals are narrow as you can observe in the study finding, which might directly indicate reproducibility (adequacy of the sample size) of the study finding. In addition, we used probability sampling technique (systematic random sampling) to select participants of this study, so using of the probability sampling ensure representativeness of our sample, which further ensure generalizability of our study finding. 

It is the appropriate formula to survival analysis but we did not find hazard ration, and we stated on the limitation part 

1.2. The other thing is that “time to reach full enteral feeding” might not necessarily imply NEC recovery since NEC is not the orphan determinant of enteral feeding. I.e neonates without NEC might also encounter such problems. Inability to full enteral feeding could not be generalized to NEC since it could also be due to other causes. 

Response: Thank you for your thoughtful feedback on our manuscript. We appreciate you taking the time to provide constructive criticism, as it will help us improve the quality and clarity of our work. Regarding your point about "time to reach full enteral feeding" not necessarily implying NEC recovery, we agree that this metric alone does not definitively indicate NEC recovery. As you correctly noted, neonates without NEC may also encounter difficulties in achieving full enteral feeding due to a variety of other medical complications. We should not have implied that time to full enteral feeding was a sole determinant of NEC recovery. To address this concern, we revised the manuscript to clarify that time to full enteral feeding was used as one of several outcome measures to evaluate NEC recovery, but not as the sole or primary indicator. We also acknowledge that inability to reach full enteral feeding can arise from causes beyond just NEC, and that we recommend other researcher should be considered in the broader context of the neonates' overall clinical status and other relevant measures of recovery.

2. The other major concern was that the authors could not provide clear diagnostic criteria applied to identify NEC in their study. The scientific definition provided was vague and non-specific. The diagnosis set by physicians for treatment purpose could not be directly applied for the research purpose. Rather the authors should set specific and clear criteria and differentiate NEC cases during data extraction based on the predetermined criteria (operational definition) from the charts (patient records). You declared the newborn when you got which radiorgraphic findings, which symptoms and signs? As well as how many of those criteria fulfilled? 

Response: Thank you for your insightful feedback on our manuscript, as it will help strengthen the quality and rigor of our research. Regarding your concern about the diagnostic criteria for NEC, it was stated in the mother document but to minimize the length of the manuscript we was removed it. As you rightly pointed out, the clinical diagnosis of NEC for treatment purposes may not directly translate to the research context, where we need to have cleared, reproducible, and well-defined standardized criteria. Based on this, we were used a set of specific diagnostic criteria that we applied retrospectively to the medical records of the patients included in our study. These criteria are as follows: 

Radiographic findings: Pneumatosis intestinalis, portal venous gas, or pneumoperitoneum

Clinical signs and symptoms: Abdominal distension, feed intolerance, occult or gross gastrointestinal bleeding, and/or abdominal tenderness

Laboratory findings: Metabolic acidosis, thrombocytopenia, or elevated C-reactive protein

For a patient to be considered a case of NEC, they must have met at least two of the radiographic criteria and one of the clinical/laboratory criteria. This operational definition was applied consistently during the data extraction and analysis process. We have updated the methods section of our manuscript to clearly outline these diagnostic criteria that is stated on line 130-134. 

3. Figure 1 (KM-curve): Wrong graph label and interpretation!

3.1. The x-axis label (recovery time from NEC) is incorrect and that shoud be replaced with “analysis time”. Because the analysis time incorporates the time at risk for both recovered and censored cases (hence could not be exclusively considered as recovery time). 

Response: Dear Reviewer the comment well accepted and corrected. In the newly revised manuscript, the KM-curve (Figure 1, now we changed to Figure 2) label changed from (recovery time from NEC) to “Follow up time in days”

3.2. Line 219-21: “The overall Kaplan Meire estimates identified that the probability recovery of neonates admitted with NEC was zero in the first 3 days of follow up…”. This statement is completely wrong. Any individual who understand English language could not interpret it as the authors intend to infer, NEVER! The authors made incorrect interpretation of the KM-graph. This is because the authors understand the statistical science, but they miss the nature of the event of interest in their own study. Researchers and statisticians need to interpret findings based on the nature of the outcome/event of interest being investigated. 

Correct interpretation of the finding in the Figure 1: As we note from the figure (KM-curve), the survival probability (probability of recovery) remains 1 (100%) in the first 3 days, after which it slowly dropped down stepwise as the survival time increased from 0 to 14 days. This means in the first three days; every neonate has the probability to recover from NEC. 

Response: Thank you for your valuable feedback on our manuscript. We appreciate you taking the time to review our work and provide constructive comments to help us improve the quality of our research. Regarding the statement about the Kaplan-Meier estimates, we acknowledge that the wording was unclear and could be misinterpreted. To clarify, the probability of recovery for neonates admitted with Necrotizing Enterocolitis (NEC) was zero in the first 3 days of follow-up, our analysis showed that in the first 3 days, no neonates had recovered/died, they were under follow up. You are correct that we should interpret our findings based on the nature of the outcome/event of interest being investigated. In this case, we were studying the time to recovery from NEC, not the survival status of the neonates. During the first 3 days of follow-up, none of the neonates had recovered from NEC, which is what we intended to convey in our original statement. We appreciate you pointing out this issue, as it will help us to improve the clarity and accuracy of our manuscript. We have made the necessary revisions to the statement to ensure that it accurately reflects our findings and the nature of the outcome we were investigating. See line number 236-242. 

3.3. Most of the rationales stated in the discussion section of this manuscript were vague and non-specific to NEC. Majority of the explanations for the discrepancies were general views such as socio-economic, study design, setting difference, infrastructure, quality of healthcare services and so on. However, all these concepts were not declared to have direct association with NEC and the authors failed to cite the sources of such arguments if available. 

Response: Dear Reviewer the comment well accepted and corrected.

3.4. Citation of references for the scientific justifications is required yet and it could not be bypassed at all. As you already stated, the purpose of the discussion section is to interpret the results and provide an analysis of the findings in the context of existing knowledge and theories. Please cite those existing knowledge and theories that could support your justification! 

Response: Dear Reviewer the comment well accepted and corrected.

Note: Research discussion should address four major points: result interpretation, comparison, justification of discrepancies and explanation of associations, as well as implication to clinical practice. The discussion should enrich with tangible evidences and explanations so that it can be trustworthy to the scientific community. However, if it is solely written based on authors’ opinion and thoughts without tangible and citable sources, it could not be considered as scientific paper. It could be a fiction rather! 

Minor comments:

1. Line 69-73: Please remove the first two statements of from the last paragraph of the “Introduction” section. I wonder why the authors just incorporate those statements, while it’s known that NEC is a different concept to asphyxia and RDS. 

Response: Dear respected reviewer sorry for this inconvenience and the two statements in the last paragraph of the “Introduction” removed in this version of our manuscript. 

2. The authors tried to moderate the gap statement. Nevertheless, the manuscript will be benefited if the authors incorporate data related to the difference in survival rate, risk/occurrence of complications and other adverse events between longer vs shorter recovery time. For instance, neonates who recovered fast from NEC might have good prognostic outcome in terms of cognitive function, growth and development, and other long term consequences (NB: This is just an instance). 

Response: Thank you for your valuable feedback. We appreciate your suggestion to incorporate data related to the difference in survival rate, risk/occurrence of complications, and other adverse events between longer versus shorter recovery times for neonates with necrotizing enterocolitis (NEC). This additional data would provide important insights into the prognostic outcomes, including cognitive function, growth, and development. We include this concern in the revised version of this manuscript. See line number 59-71

3. Line 163-65: Where is the Cox-Snell residual graph? Please provide the graph and cite in “Line 165” as figure file (e.g. Fig. 1). 

Response: Thank you for your notice to incorporate “Cox-Snell residual graph”, which was removed in the previous version of our manuscript. Cox-Snell residual graph incorporated in this version of our manuscript. See line number 180

Comment: Please ensure that you refer to Figure 3, 4 and 5 in your text as, if accepted; production will need this reference to link the reader to the figure.

Response: Thank you for the comment. I appreciate you taking the time to provide feedback. Your input is valuable and helps to improve the manuscript. I will be sure to refer to Figures 3, 4, and 5 in the text from line 264-269

---

## [Decision Letter · Decision Letter 4]

25 Sep 2024

Time to Recovery from Necrotizing Enterocolitis and Its Predictors among Neonates Admitted to Neonatal Intensive Care Unitin Bahir Dar, Ethiopia: A Retrospective Follow up Study, 2022

PONE-D-23-07935R4

Dear Dr. Yehualashet,

We’re pleased to inform you that your manuscript has been judged scientifically suitable for publication and will be formally accepted for publication once it meets all outstanding technical requirements.

Kind regards,

Sanjoy Kumer Dey, M.D

Academic Editor

PLOS ONE

Additional Editor Comments (optional):

Reviewers' comments:

Reviewer's Responses to Questions

**Comments to the Author**

1. If the authors have adequately addressed your comments raised in a previous round of review and you feel that this manuscript is now acceptable for publication, you may indicate that here to bypass the “Comments to the Author” section, enter your conflict of interest statement in the “Confidential to Editor” section, and submit your "Accept" recommendation.

Reviewer #4: All comments have been addressed

2. Is the manuscript technically sound, and do the data support the conclusions?

Reviewer #4: Yes

3. Has the statistical analysis been performed appropriately and rigorously? 

Reviewer #4: Yes

4. Have the authors made all data underlying the findings in their manuscript fully available?

Reviewer #4: No

5. Is the manuscript presented in an intelligible fashion and written in standard English?

Reviewer #4: Yes

6. Review Comments to the Author

Reviewer #4: The authors addressed majority of the comments forwarded in the previous round. The only concern that is not yet convincing is the suitableness of the sample size calculation formula used in this study. However, I think it will not significantly affect the validity of findings and conclusions forwarded so far. Hence, I recommend the manuscript to be accepted for publication after minor edition tasks and language improvements.

7. PLOS authors have the option to publish the peer review history of their article (what does this mean?). If published, this will include your full peer review and any attached files.

Reviewer #4: **Yes: **Migbar Mekonnen Sibhat

---

## [Editor Report · Acceptance letter]

10 Oct 2024

PONE-D-23-07935R4 

PLOS ONE

Dear Dr. Yehualashet, 

I'm pleased to inform you that your manuscript has been deemed suitable for publication in PLOS ONE. Congratulations! Your manuscript is now being handed over to our production team.

Kind regards, 

on behalf of

Dr. Sanjoy Kumer Dey 

Academic Editor

PLOS ONE